# Rapid fabrication of gold microsphere arrays with stable deep-pressing anisotropic conductivity for advanced packaging

An Cao[1,7], Yi Gong[1,2,7], Dilong Liu [1,7] ✉, Fan Yang[3], Yulong Fan [4,5,6] ✉,
Yinghui Guo[4,5], Xingyou Tian[1] & Yue Li [1,3] ✉

Smooth metal microspheres with uniform sizes are ideal for constructing particle-arrayed anisotropic conductive films (ACF), but synthesis is hindered by challenges in controlling anisotropic metal growth. Here, we present a positioned transient-emulsion self-assembly and laser-irradiation strategy to fabricate pure gold microsphere arrays with smooth surfaces and uniform sizes. The fabrication involves assembling gold nanoparticles into uniform colloidosomes within a pre-designed microhole array, followed by rapid transformation into well-defined microspheres through laser heating. The gold nanoparticles melt and merge in a layer-by-layer manner due to the finite skin depth of the laser, leading to a localized photothermal effect. This strategy circumvents anisotropic growth, enables tunable control of microsphere size and positioning, and is compatible with conventional lithography. Importantly, these pure gold microspheres exhibit stable conductivity under deep compression, offering promising applications in soldering micro-sized chips onto integrated circuits.

Anisotropic conductive adhesive film (ACF) is recognized for its distinctive Z-axis conductivity while ensuring insulation in X-Y plane. This feature enables the vertical compact bonding of electronic components, advancing device density in integrated packaging[1–5]. Early ACF versions used sparsely distributed conductive microspheres in a binder matrix, serving and simplifying vertical circuit interconnection[6,7]. However, as circuits shrunk and densified in resolution, traditional ACF approached its limits in preventing lateral circuit leakage caused by dislocations and jamming of particles. A breakthrough solution emerged: pre-patterning of the adhesive film with micro-sized intaglio arrays (such as holes) and placing one conductive bead in each hole, known as particle-arrayed ACF[8,9]. This ensures precise and reliable conductivity, providing an essential solution for high-resolution

display components laminating on integrated circuits (IC), such as Micro Light Emitting Diode (μLED) chips. While, implementing this strategy demands strict size uniformity of the microspheres and accurate microsphere arrangement with nearly 100% filling rate on an ultra-large scale[10]. Therefore, achieving the precision in microsphere synthesis and positioning is crucial for developing particle-arrayed ACF products, presenting a promising avenue for wafer-scale bonding of high-resolution terminals.

Thanks to advanced polymerization techniques, the batch synthesis or screening of polymer microspheres with ultra-uniform size (<5% in coefficient of variation) has become feasible[11–13]. Plating these polymer microspheres with a thin layer (~100 nm in thickness) of metal materials (like gold or nickel) endows them with reliable

[1]Key Lab of Materials Physics, Anhui Key Lab of Nanomaterials and Nanotechnology, Institute of Solid State Physics, HFIPS, Chinese Academy of Sciences, Hefei, PR China. [2]China-Europe Electronic Materials International Innovation Center, Hefei, PR China. [3]Tiangong University, Tianjin, PR China. [4]National Key Laboratory of Optical Field Manipulation Science and Technology, Chinese Academy of Sciences, Chengdu, PR China. [5]State Key Laboratory of Optical Technologies on Nano-Fabrication and Micro-Engineering, Institute of Optics and Electronics, Chinese Academy of Sciences, Chengdu, PR China. [6]Quantum Science Center of Guangdong-HongKong-Macao Greater Bay Area (Guangdong), Shenzhen, PR China. [7]These authors contributed equally: An Cao, Yi Gong, Dilong Liu. ✉e-mail: dlliu@issp.ac.cn; yulong_fan2017@163.com; yueli@issp.ac.cn

conductivity, known as polymer electroless plating[14–16], which has been widely employed in industrial fabrication. However, the wet-chemical approach to metal plating exhibit anisotropic growth due to intrinsic energy differences in crystalline faces, resulting in a relatively rough metal surface on the polymer microsphere. Furthermore, the weak chemical bond between the metal shell and polymer core poses limits during the lamination under high-pressing stress, often leading to metal shell cracking or detachment and compromising overall bond conductivity. These issues intensify as the sphere size decreases (<2 μm), necessitating stricter size uniformity requirements[17]. In contrast, direct seed growth of pure gold microspheres with uniform size and ultra-smooth morphology might address these challenges. Their inherent ductility and malleability ensure superior conductivities especially under deep-pressing stress, making them ideal for ACF application. However, chemical growth approaches still adhere to the metallic anisotropic growth principle, leading to the formation of polyhedral structures[18]. Despite recent achievements, such as the slow "growth-etching" cycling method reported by Lee et al.[19], which produces ultra-smooth and highly-spherical gold particles, achieving smooth pure microspheres larger than 200 nm remains elusive. Therefore, substantial challenges persist in fabricating uniform, smooth and pure metal microspheres, primarily stemming from the fundamental bottleneck of anisotropic metal growth during synthesis.

Beside microsphere synthesis, the second essential step is the accurate positioning of these microspheres into pre-patterned microhole arrays[20,21]. Since most conductive microspheres were synthesized and well-dispersed in solution, template-assisted self-assembly strategies driven by capillary forces of solvent were commonly used for placing colloidal-state microspheres into microholes[22–25]. However, as template-assisted capillary self-assembly relies on delicate evaporation of the microsphere suspension, it is prone to random particle aggregations due to uncontrollable three-phase line retraction, making long-range accuracy in positioning difficult[24,26–29]. Mechanical rubbing of microsphere powders is a solvent-free strategy for quick, large-scale and high-filling-yield positioning of the microsphere into microhole patterns[8]. While undirected squeeze of these microspheres under strong mechanical forces will cause an irreversible deformation and detachment of the metal coating, thereby destroying the final conductivity[30]. Therefore, to realize the fabrication of particle-arrayed ACF products, exploring a novel strategy that not only can overcome the anisotropic growth issues in metal but also address the accurate and intact positioning challenges becomes imperative.

The nanosecond-laser pulse irradiation technique is well-known for rapidly heating and reshaping gold nanostructures into near-perfect nanospheres, effectively avoiding the anisotropic metal growth[31,32]. Notably, under such laser irradiation, small clusters of assembled gold nanoparticles can be fused into a larger sphere[33,34]. Inspired by this fusion process, if we assemble gold nanoparticles into uniform large clusters or colloidosomes and then expose to laser irradiation, it could achieve the rapid production of pure gold microspheres without the issues of anisotropic growth. Fortunately, our group previously reported a unique transient emulsion system based on partial miscibility of water and 1-butanol, which enables the water phase precisely emulsified in each microhole pre-patterned on substrate, allowing positioned self-assembly of dispersed nanoparticles into uniform colloidosomes and confined inside, termed as positioned transient-emulsion self-assembly technique[35]. Herein, based on such positioned self-assembly followed with a laser-irradiated ripening process, we developed a simple, rapid and efficient strategy to fabricate uniform, smooth pure gold microsphere array accurately positioned in the pre-patterned microholes on substrate, rapidly forming a new kind of particle-arrayed ACFs within 1 min. This strategy would refresh the inertia "synthesis first and then positioning" conception in the fabrication of ACF products, which is robust and suitable for high-resolution electronic-interconnection applications.

## Results
### The positioned self-assembly and laser-irradiated ripening strategy

Figure 1a illustrates the process of fabricating pure gold microsphere arrays through a positioned self-assembly and laser-irradiated ripening strategy. Typically, aqueous colloidal gold nanoparticles with $60 \pm 3.4$ nm in diameter were used as the gold source for self-assembly (Supplementary Fig. 1). Hydrophobic polystyrene (PS) substrates patterned with a honeycomb microhole array (4.5 μm in diameter, 4.1 μm in depth and 6.0 μm in periodicity) were employed as the positioning template (Supplementary Fig. 2). First of all, the gold nanoparticles were assembled into uniform colloidosomes confined in templating holes based on a water-in-butanol transient emulsion system, as reported in our previous works[35,36]. Briefly, the microhole-array substrate wetted with 1-butanol was added with a sufficient amount of aqueous gold nanoparticle solution. The partial miscibility of 1-butanol and water causes the aqueous gold nanoparticle solution to occupy the microholes. A large amount of 1-butonal solvent was then employed to wash away the surplus aqueous phase, emulsifying colloidal solution into uniform emulsion droplets within the microholes and driving self-assembly of gold nanoparticles into colloidosomes through water diffusion into 1-butanol. Figure 1b and Supplementary Fig. 3 present typical scanning electron microscope (SEM) images of the assembled gold colloidosomes based on this self-assembly approach. These gold colloidosomes demonstrate a narrow size distribution at $705 \pm 32.9$ nm in average and accurately confined in the templating microhole array.

To achieve laser-induced ripening, a nanosecond pulse laser with a wavelength of 532 nm was selected to match the intrinsic absorption peak of gold nanoparticles (Supplementary Fig. 1a). When exposed to multiple laser pulses, the colloidosomes efficiently absorbed pulse energy, rapidly ripening into micro-sized melted-state droplets due to the photo-thermal effect from their localized surface plasmonic resonance and adjacent plasmonic coupling properties[31,37,38]. In most case, the ripening process follows a layer-by-layer melting and merging mechanism in the morphology evolution, where the heating-melting of gold nanoparticles in colloidosomes progresses layer-by-layer, as illustrated in Fig. 1c. This mechanism was observed experimentally by controlling the number of laser pulses while maintaining a fixed laser fluence at 22.9 mJ·cm$^{-2}$, which is easy to reach the melting point region of gold (1337 K in theory). During laser irradiation, only the top-surface layer of gold nanoparticles in colloidosomes was exposed, leading to a rapid heating, melting fusion into slightly larger gold droplets, as confirmed by the second SEM image in Fig. 1c when only a single laser pulse was applied (Supplementary Fig. 4a). Following the same principle, with an increase in the number of laser pulses, the remaining nanoparticles underwent a heating-melting process layer-by-layer, and adjacent ripened droplets spontaneously merged to minimize the free energy of system. As observed in the SEM results in Fig. 1c and Supplementary Fig. 4b–d, increasing the number of laser pulses from 10 to 50 resulted in clear evidence of inner gold nanoparticle melting and ripened droplet merging. After 50 pulses, all the gold nanoparticles were melted and merged into a single micro-sized sphere, and further increasing of the pulse number barely changes the morphology of microspheres (Fig. 1c). This indicates the feasibility of moving the laser around to achieve uniform and large-scale fabrication of gold microsphere arrays without affecting the formed gold microspheres (Supplementary Fig. 5). Slightly cooling them below the melting point, also termed as quenching, suppressed the reorientation of lattice gold atoms and maintained the spherical morphology of droplets, leading to the formation of pure gold microsphere arrays (Fig. 1d). Corresponding size statistics reveal the average diameter of the gold microspheres in Fig. 1d is $557 \pm 20.4$ nm with a coefficient of variation of 4.03%, as outlined in Fig. 1e and Supplementary Fig. 6, indicating an excellent uniform size distribution of microspheres prepared by this

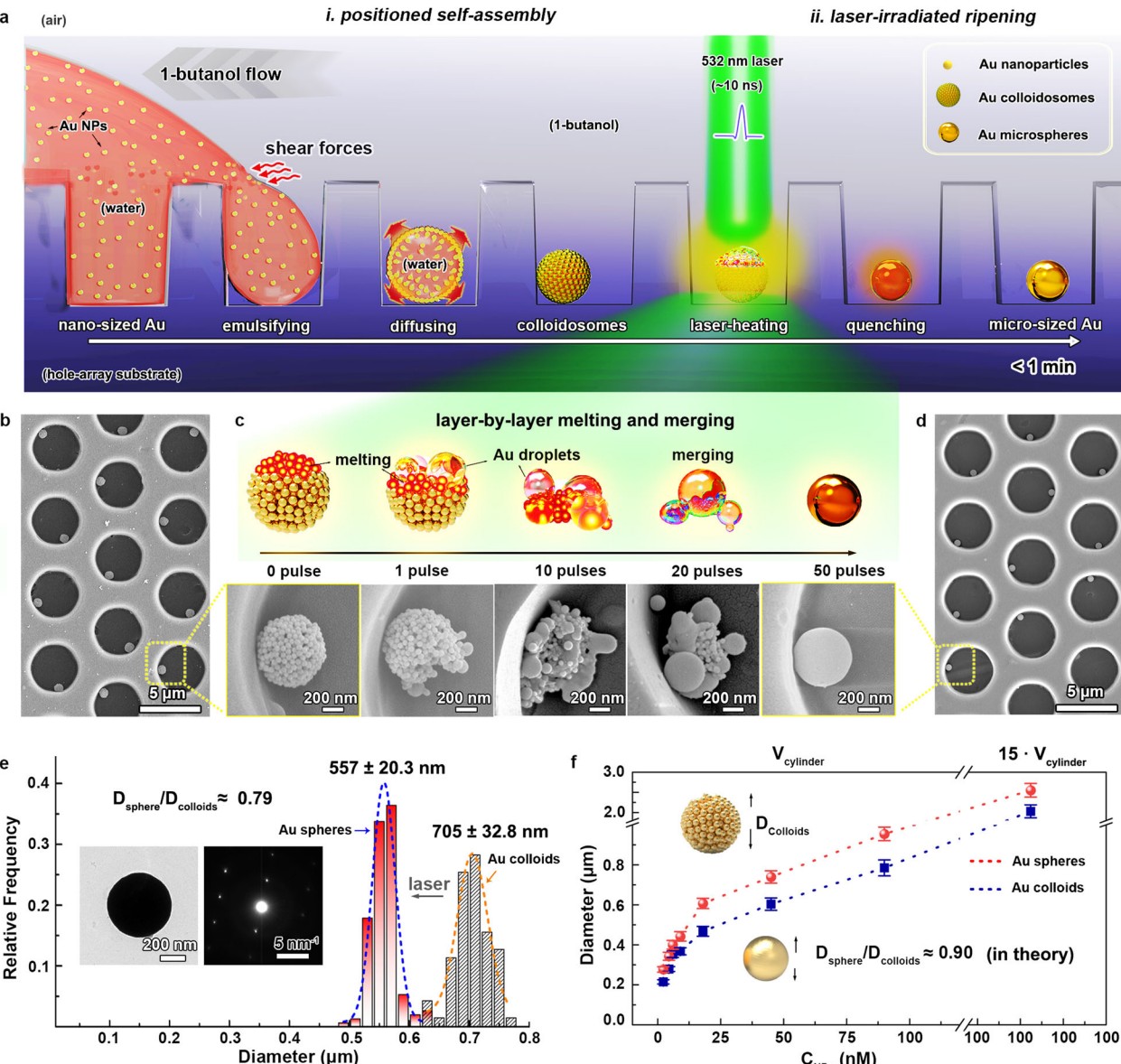

**Fig. 1 | A rapid approach to pure gold microsphere array fabrication.**
**a** Schematic illustration of the well-positioned transient-emulsion self-assembly and laser-irradiated ripening strategy for pure gold microsphere array fabrication, where the dark-blue background indicates the 1-butanol solvent environment. **b** A typical SEM image of gold colloidosomes obtained by the positioned transient-emulsion self-assembly. **c** Layer-by-layer melting and merging evolution of gold colloidosomes into gold microsphere under laser irradiation. **d** Corresponding SEM image of pure gold microsphere array after the laser-irradiated ripening process. **e** Diameter statistics of gold microspheres and colloidosomes, insets showing typical TEM image and SAED pattern of a gold microsphere. **f** Diameter control of gold colloidosomes and microspheres stemmed from adjusting the concentrations of aqueous gold nanosphere solution and the volume of the microholes. The error bars were obtained based on at least 50 values. Source data are provided as a Source Data file.

strategy. Transmission electron microscope (TEM) image of a typical gold microsphere reveals its near-perfect spherical shape and smooth surface, and its selected area electron diffraction (SAED) pattern demonstrates its single-crystalline feature (insets in Fig. 1e).

Despite the laser-induced high temperatures in melting the gold colloidosomes, the template materials remain unaffected due to the short time in irradiation (~30 ns), confirming the key role of the photo-thermal effect of gold nanoparticles in this strategy. Notably, the entire positioned self-assembly and laser-ripening process was completed within 1 min, making it ideal for industrial fabrication. Moreover, a significant advantage of this strategy lies in the size distribution of the obtained gold microspheres determined by the templating microhole arrays, simplifying control through conventional lithography techniques. Besides, adjusting the concentration of the aqueous gold nanoparticle solution within the templating microholes allows further tuning the colloidosome size. Figure 1f summarized the relationships among employed gold nanoparticle concentrations, diameters of assembled gold colloidosomes and irradiated gold microspheres. As gold nanoparticle concentration increases within a fixed templating microhole volume ($V_{cylinder}$), colloidosomes grow from 0.25 to 0.96 μm in diameter (red dash line in Fig. 1f), followed with a slight decrease in diameter for gold microsphere (0.2–0.79 μm, see blue dash line in Fig. 1f) after laser irradiation. With a 15-fold increase in the templating hole volume, colloidosomes can reach 2.5 μm and the resulting gold microsphere can grow to 2.0 μm (the laser fluence was increased to 39.1 mJ·cm$^{-2}$). The diameter ratio between the irradiated microsphere and the initial colloidosomes ($D_{sphere}/D_{colloids}$) is around 0.79–0.81, approximating the theoretical value of 0.90 in a hexagonal-close-packed model (detailed calculations in Supplementary Note 1). Any deviation should be attributed to the dislocations, defects, and non-

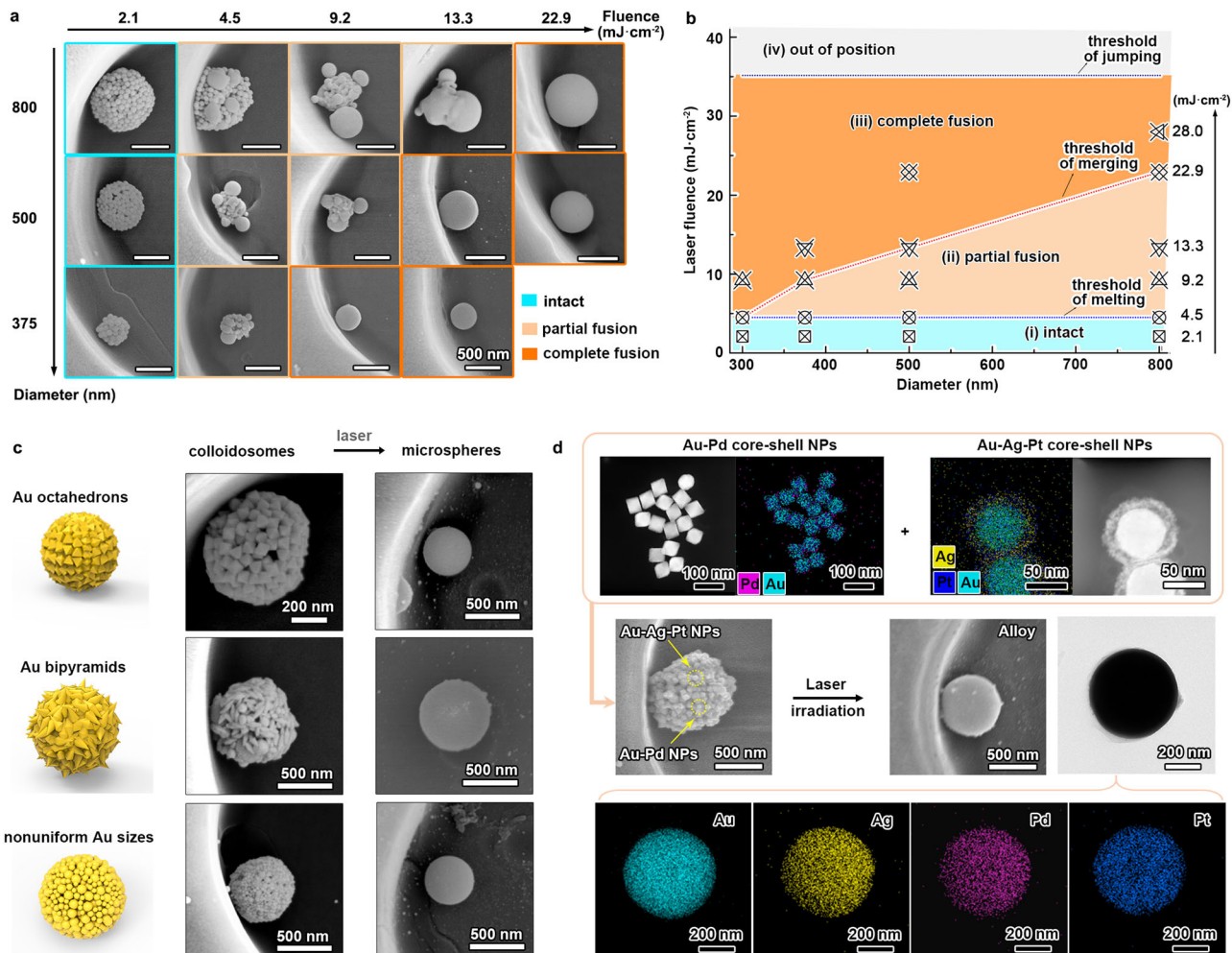

**Fig. 2 | Laser fluence factor and versatility. a** Typical SEM images of gold colloidosomes with diameters of 375, 500, and 800 nm irradiated under varied laser fluences. **b** A phase diagram of the ripening evolution of gold colloidosomes under varying laser fluences according to the results of (**a**), the marks represent the position of the experimental results on the phase diagram. **c** Typical SEM images of Gold colloidosomes assembled from gold octahedrons, bipyramids, and non-uniform nanoparticles and their corresponding ripened gold microspheres after laser irradiation. **d** Typical scanning-TEM images of Au-Pd and Au-Ag-Pt core-shell nanoparticles used as building blocks, SEM images of their assembled colloidosomes and ripened Au-Ag-Pd-Pt alloy microspheres after laser irradiation, and EDS mapping of these four elements. Source data are provided as a Source Data file.

close-packing of gold nanoparticles. Therefore, a simple, rapid, and straightforward approach was presented to fabricate pure gold microsphere arrays, based on the positioned transient-emulsion self-assembly of gold nanoparticles followed with a laser-irradiated ripening process. A comprehensive comparison with other techniques for fabricating nano- and micro-sized gold spheres are also summarized in Supplementary Table 1.

### The laser fluence factor in gold microsphere fabrication

The successful ripening of gold colloidosomes into perfect and smooth gold microspheres relies on a crucial parameter of laser: laser fluence. To reveal the impact of laser fluence on the ripening process, we employed gold colloidosomes with diameters of 375, 500, and 800 nm to expose on varied laser fluences, as presented in Fig. 2a. When the fluence is relatively low, like 2.1 mJ·cm$^{-2}$, it barely melted the gold nanoparticles, leaving the colloidosomes intact. A slight increase to 4.5 mJ·cm$^{-2}$ induced melting and fusion of the top-surface gold nanoparticles, resulting in a partial-fusion state for all sizes of gold colloidosomes. It is noteworthy that extending the pulse numbers or irradiation times did not improve this partial-fusion state. Further increasing the fluence to 9.2 mJ·cm$^{-2}$ triggered more inner gold nanoparticles to melt, leading to the merging growth of ripened

spheres or even achieving complete fusion into single smooth microspheres, as observed for the smallest colloidosomes. A similar trend continued at 13.3 mJ·cm$^{-2}$, where already complete-fusion microspheres remained unaffected. All colloidosomes reached a complete fusion state at 22.9 mJ·cm$^{-2}$. At higher laser fluence (like 30.0 mJ·cm$^{-2}$), there was minimal impact on the morphology of already-ripened microspheres, but they could jump out of microhole positions due to excessive pulse pressure (Supplementary Fig. 8a). Extremely high laser fluence (over 50 mJ·cm$^{-2}$ reaching the boiling point region at 3246 K in theory) induced a boiling-to-evaporation process of gold nanoparticles, resulting in the formation of a serial of ultra-small and randomly-sized nanospheres (Supplementary Fig. 8b).

Based on the results from Fig. 2a, we drafted a phase diagram illustrating the ripening evolution of colloidosomes with varying laser fluences, depicting four states: intact, partial fusion, complete fusion, and out of position, as shown in Fig. 2b. Three laser fluence thresholds for state transitions were identified: melting in nanoparticles, merging into single microsphere, and jumping out of position. The thresholds for melting and jumping remained relatively constant, dictated by similar photo-thermal effects and fluid dynamics related to initial nanoparticle size and surrounding environment. In contrast, the merging threshold increased with colloidosome diameter, as larger

gold particles required higher laser fluence for melting during the merging growth of intermediately-ripened microspheres. This aligns with previous research[39], where the theoretical relationship between laser pulse fluence ($J(d_p)$) and diameter of gold microspheres ($d_p$) was simplified into: $J(d_p) \propto d_p^2$, with detailed equation elaborations provided in the methods. In short, the threshold for merging is typically determined by the largest gold nanoparticles during the ripening process. Additionally, since the laser-heating (~30 ns) and quenching (1–10 μs) time are much shorter than the interval (0.05 s) between two pulses, the laser energies of two pulses cannot be cumulatively absorbed[40,41]. Consequently, increasing the pulse number below the merging threshold does not result in the formation of completely-fused microspheres, consistent well with experimental observations.

## Versatility in fabricating pure gold and alloy microspheres

This positioned self-assembly and laser-irradiated ripening strategy is highly versatile, allowing the use of various plasmonic nanostructures as building blocks, regardless of their shapes, sizes, or components, as long as they exhibit strong photo-thermal effects under laser pulse irradiation. As exemplified in Fig. 2c, gold octahedrons, bipyramids, and nonuniform gold nanoparticles can all be assembled into uniform gold colloidosomes, subsequently ripening into gold microspheres upon laser irradiation (Supplementary Figs. 9–12). It is worth mentioning that the building block size and shape affects the minimum laser fluence needed to initiate the melting process due to absorption differences, but it does not impact the final transformation of colloidosomes into microspheres (Supplementary Fig. 9). Beyond pure gold microspheres, this strategy also facilitates the production of alloy microspheres. For instance, employing Au-Pd and Au-Ag-Pt core-shell nanoparticles as building blocks in colloidosome assembly enables the straightforward production of smooth Au-Ag-Pd-Pt alloy microspheres upon laser irradiation, as presented in Fig. 2d. All the above samples were irradiated with laser pulses at a fluence of 22.9 mJ·cm$^{-2}$ and a frequency of 10 Hz for a duration of 10 seconds. Energy dispersive spectroscopy (EDS) mappings confirm the even distribution of these four elements in the alloy microspheres, attributed to the high temperature generated by the laser irradiation[42,43]. The robustness of this strategy holds promise in providing an alternative way for the rapid fabrication of uniform-sized high-entropy alloys[44].

## Simulating laser interactions with gold colloidosomes

The transformation of colloidosomes into microspheres under laser irradiation was described as "ripening" in this context, while it involves a mechanism totally distinct from well-known "Ostwald ripening"[45]. Instead, it relies on laser-induced plasmonic heating, melting fusion and merging growth, which has been widely used for reshaping colloidal nanostructures and their clusters into perfect nanospheres[40,41,46]. In the case of complex gold colloidosomes, the melting and merging process progressed in a unique layer-by-layer way. To comprehend the underlying mechanism of this phenomenon, we first conducted full-wave simulations to quantify the optical behavior of plasmonic structures, and then investigated the thermal dynamics of the heating structures to reveal how laser pulses heat up the gold colloidosomes. This optical-thermal calculation method has been successfully adopted before to elucidate the plasmo-electric potential effect[47,48], blackbody-like gold nanostructures for high-efficiency photothermal therapy[40,49], heating induced strain relaxation in solar cell[50] and heating accelerated pyro-catalytic chemical reaction[51].

We created colloidosome models with diameters of 399, 509, and 684 nm, comprising 60-nm gold nanospheres arranged in a hexagonal-close-packed configuration for simulations. The colloidosomes models were positioned in a polymer hole substrate and the surrounding media was air for simplification. Using the finite difference time domain (FDTD) method, we simulated the cross-sectional electromagnetic field distributions of these gold colloidosomes, revealing

their complex optical properties—absorption, Mie scattering, and extinction spectra (Fig. 3a). All the gold colloidosomes exhibit broadened absorption cross section as compared to Mie scattering nanostructures[52,53], due to the multiple scattering effect. The choice of the intrinsic absorption peak around 530 nm as laser wavelength for melting process facilitates the experiment with colloidosomes of different sizes, without the need to change the laser's wavelength according to the shift of colloidosomes' absorption peaks in the near-infrared regime. Insets in Fig. 3a present the cross-sectional electromagnetic field distributions of colloidosomes excited at 532 nm. The enhanced electromagnetic field was localized in the gaps between surface gold nanoparticles, owing to strong near-field plasmonic coupling[54–56]. Larger colloidosomes exhibited more hotspots on the surface of colloidosomes, resulting in larger absorption and scattering cross-sections (Fig. 3b). This can be attributed to more surface gold nanoparticles exposed under irradiation. Interestingly, the gold colloidosome exhibits quasi linear lineshapes of both the absorption cross section and scattering cross section with respect to its diameter. This results in the quadratic dependence of $J(d_p)$ on $d_p$ (as discussed in the Supplementary Note 2), leading to the requirement of higher laser fluence for effective heating to melt the gold colloidosomes, consistent with experimental results (Fig. 2a, b). Importantly, comparing the size dependent absorption cross sections of gold colloidosomes and pure gold nanospheres (300, 375, and 500 nm, see Supplementary Fig. 14 for more simulation details), we can find that $K_{absorption}$ of the former is much lager than the latter one (Fig. 3c), which implies that the former is always more efficient in energy conversion for melting. Besides, the absorption efficiency (defined as the ratio of absorption cross section to the extinction cross section) of gold colloidosomes is always higher and more stable with the increase of colloidosome's size, indicating the more efficient optical energy utilization than the pure gold nanospheres (Supplementary Fig. 15).

Further utilizing the commercial finite-element-method software COMSOL 5.5 Multiphysics method, heat transfer module, we simulated the laser-induced photo-thermal response of gold colloidosomes exposed to a nanosecond 532-nm laser pulse. The skin depth at this wavelength was determined to be ~38.7 nm, implying that the laser effectively penetrates only the surface layer of nanoparticles (Fig. 3d). This surface layer of nanoparticles, termed the nanoparticle Union (inset in Fig. 3d), was distinctly observed from inner colloidosomes. Laser fluences of 9.2, 13.3, and 17.3 mJ·cm$^{-2}$ (Fig. 3e) were applied to gold colloidosomes with diameters of 399, 509, and 684 nm, respectively. The entire temperature of the system, primarily focusing on Union, inner colloidosomes, and the polymer substrate, were monitored during irradiation by a single laser pulse (Fig. 3f). It is clear to see the temperature of Union rapidly exceeds the melting point of crystalline gold nanoparticles (1337 K in theory[57]) within 38 ns, even under a low laser fluence of 9.2 mJ·cm$^{-2}$. In contrast, the average temperature of inner colloidosomes remains significantly below the melting temperature, even under a much higher laser fluence (17.3 mJ·cm$^{-2}$). The polymer substrate constantly maintains room temperature throughout this irradiation process. Insets of Fig. 3f present typical cross-sectional temperature field distributions of these colloidosomes at their peak temperatures. It can be directly observed only the surface layer of nanoparticles is heated to the melting point, initiating the fusion of the surface layer of nanoparticle and exposing more inner gold nanoparticles. Analogously, with an increasing number of laser pulses, these colloidosomes undergo a layer-by-layer process of heating, melting and merging until they successfully ripen into gold microsphere at an optimal laser fluence. These simulation results are agreed well with experimental findings in Fig. 1c. In contrast, the temperature rises of the gold microspheres transformed from gold colloidosomes can only reach 100–200 K during one pulse illumination due to the high thermal conductivity resultant quasi-isothermal inside gold (Supplementary Fig. 16), highlighting the advantage of the

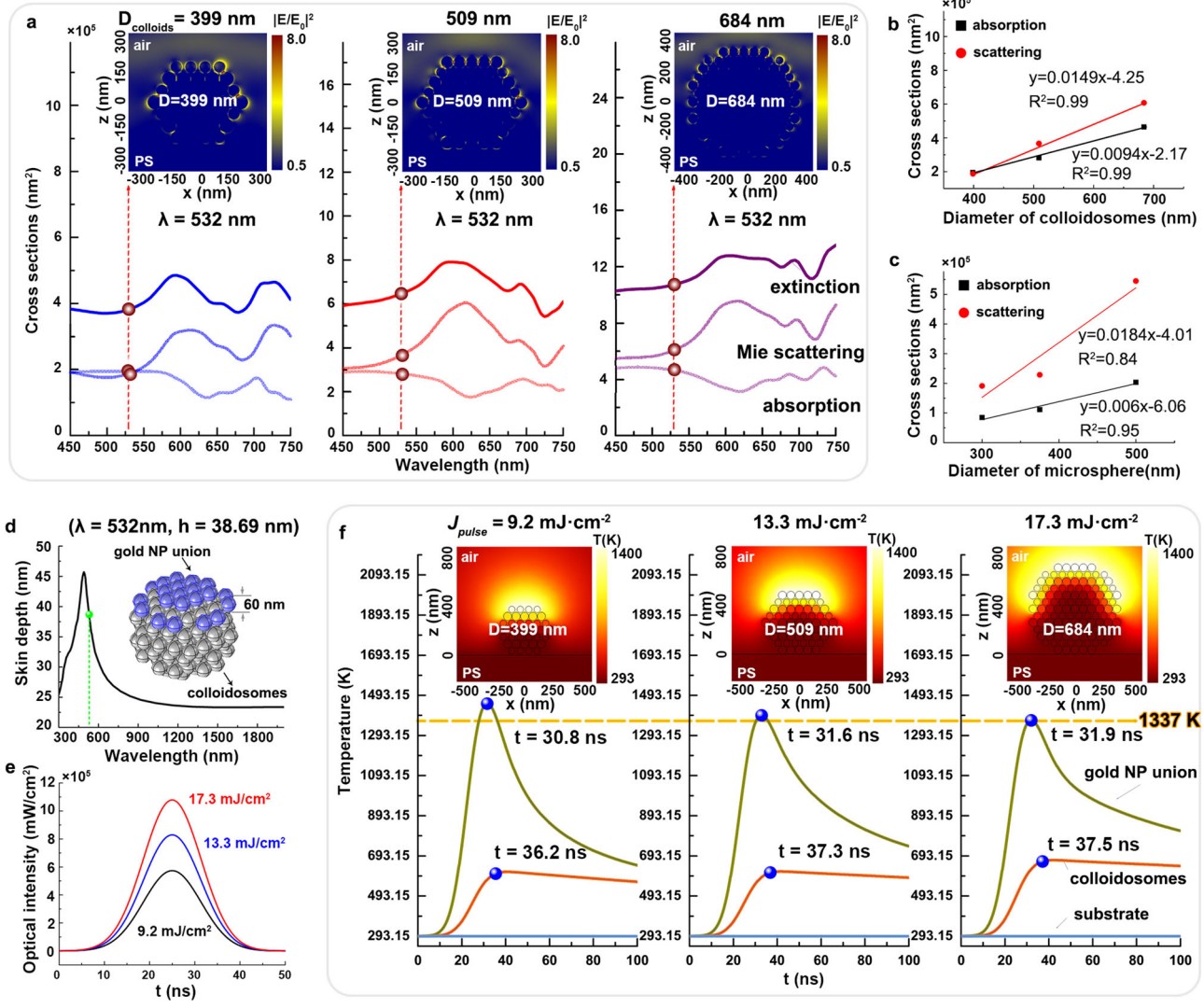

**Fig. 3 | Simulation of laser interactions with gold colloidosomes. a** Simulated cross-sectional electromagnetic field distribution of gold colloidosomes with diameters of 399, 509, and 684 nm under circularly-polarized light illumination, providing their optical properties—absorption, scattering, extinction spectra based on the FDTD method. Depictions of the variation trends of the absorption and scattering cross sections of gold colloidosomes (**b**) and pure gold microspheres (**c**) with varying diameters excited at a laser wavelength of 532 nm. The fitting equations are all y = a + b × x, (**d**) Skin depth simulation as a function of optical wavelength, inset presented a 3D model of gold colloidosomes for simulations. **e** The typical optical intensity of laser pulse with fluences of 9.2, 13.3, and 17.3 mJ·cm$^{-2}$. **f** Simulated temperature evolution of the system over time during laser pulse irradiation, focusing on Union, inner colloidosomes and polymer substrate. Insets show typical cross-sectional temperature field distributions of gold colloidosomes at their peak temperatures. Source data are provided as a Source Data file.

heating, melting and merging process of gold colloidosomes. As a result, based on our above analysis, we believe that using gold colloidosomes as heat source demonstrates at least three folds of advantages, such as larger absorption cross section, higher energy conversion, and higher surface temperature.

## Heat dissipation during laser irradiation

The photo-thermal response of gold colloidosomes to laser irradiation varies when the environment changes from air to liquid. For instance, in Fig. 4a, we present the cross-sectional electromagnetic filed distribution of 399-nm colloidosomes in 1-butanol at wavelength of 532 nm, along with the temperature field distribution under laser irradiation at 9.2 mJ·cm$^{-2}$. Comparative analysis of absorption, scattering and extinction cross-sections for such colloidosomes in both air and 1-butanol is illustrated in Fig. 4b (see Supplementary Figs. 14 and 17 for additional details). It can be found the electromagnetic field distribution and optical properties appear similar in both environments. However, when exposed to laser, the elevated temperature of gold

colloidosomes in 1-butanol is lower than in air (Fig. 4c and Supplementary Fig. 18). This temperature reduction is attributed to the distinct thermal conductivity and heat capacity of the surrounding medium. In comparison to air, 1-butanol has a higher thermal conductivity and a higher heat capacity (see thermal properties used in simulation in "Simulation method"), facilitating heat transfer from gold to environment with lower temperature rise[58]. Further monitoring media temperature in simulations performed by COMSOL during the laser irradiation confirms the steady temperature in 1-butanol while the air experiences heating, providing direct evidence (Fig. 4d). Importantly, despite the more efficient transformation of gold colloidosomes into gold microspheres in air (requiring fewer laser pulses, even just one) due to its higher heat insulation, the rapid transformation induces a higher deformation force[59,60], causing gold microspheres to jump out of position from the microholes (Fig. 4e). In contrast, gold microspheres in a liquid environment (e.g., 1-butanol) encounter fluidic resistance and generate relatively lower deformation forces, preventing them jumping out of the microholes. After multiple

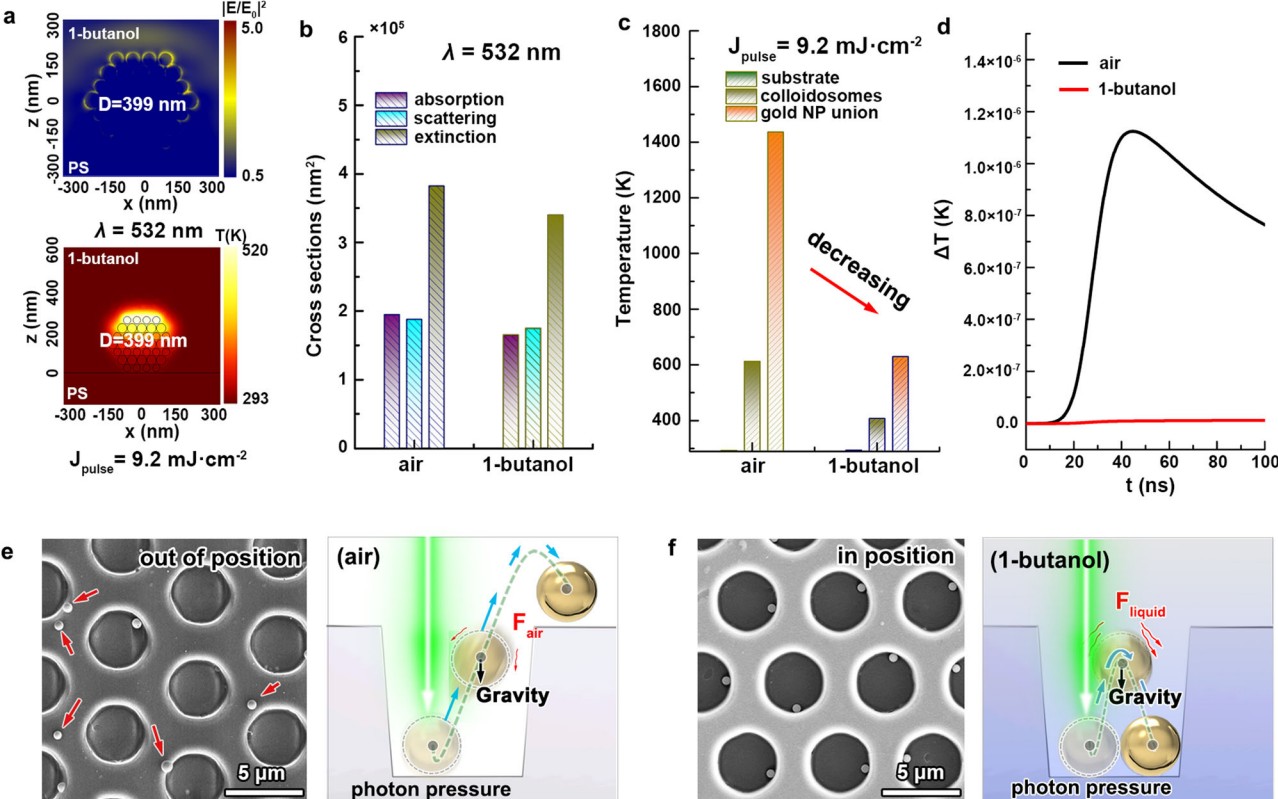

**Fig. 4 | Heat dissipation during laser irradiation. a** The cross-sectional distribution of electromagnetic field for colloidosomes in butanol (top), along with the temperature distribution under irradiation (bottom). Comparison analysis of the optical properties (**b**) and the elevated temperature (**c**) of gold colloidosomes in 1-butanol versus in air. **d** The environment temperature monitoring based on the laser irradiations, they usually stay in their initial microholes, forming an in-position state of a microsphere array (Fig. 4f). Hence, despite the introduction of liquid during laser irradiation leading to a lower temperature, it effectively preserves the positioned state of formed microspheres. It is noteworthy that when the laser fluence is too high (Fig. 2b), the deformation forces can be strong enough to overcome the fluidic resistance, causing the microspheres to jump out of position again. COMSOL simulation. Typical SEM images and schematic illustrations of the laser-induced jumping behavior of the gold microspheres in air (**e**) and in 1-butanol (**f**), demonstrating an out-of-position and in-position state, respectively. Source data are provided as a Source Data file.

laser irradiations, they usually stay in their initial microholes, forming an in-position state of a microsphere array (Fig. 4f). Hence, despite the introduction of liquid during laser irradiation leading to a lower temperature, it effectively preserves the positioned state of formed microspheres. It is noteworthy that when the laser fluence is too high (Fig. 2b), the deformation forces can be strong enough to overcome the fluidic resistance, causing the microspheres to jump out of position again.

## Stable conductivity of pure gold microspheres under deep compression

Nano-indentation system was employed to assess the conductive durability of pure gold microspheres under deep compression, as depicted in Fig. 5a. Testing particles, including 5 μm commercially polymer-plated gold microspheres (top right) and 2 μm pure gold microsphere prepared through our laser irradiation strategy (bottom right, more details can be seen in the Supplementary Fig. 19), were sparsely deposited on a flat tungsten plate. Conductivity properties were probed using a flat tungsten carbide indenter ($100 \times 100$ μm²) with programmable displacement. Specially, the displacement program was set to comprise three compression stages: slight press (0–10 s), deep press (10–30 s), and retraction (30–60 s), executed at a load rate of 100 mN/min with a maximum load of 50 mN (Fig. 5b). For plated gold microsphere, its conductivity exhibited rapid electrical "on" during slight press, but quickly turned "off" during deep press, as indicated by the red curve in Fig. 4c. This abrupt "turning off" around 10 s resulted from irreversible cracking and detachment of the gold shell from the polymer core. This was confirmed from the retraction of compression force, where conductivity did not recover, and the

corresponding SEM image capturing the cracked plated-gold microsphere after deep compression (inset in Fig. 4c). In contrast, pure gold microsphere displayed consistently stable "on" conductivity throughout the entire compression process and even the retraction process (Fig. 4d). This enduring conductivity is attributed to the superior ductility and malleability of pure gold materials in response to deformation. The corresponding SEM image exhibits a round-pie-like gold plate of the microsphere after deep compression, confirming its uniform deformation (inset in Fig. 4d). Such pure gold microspheres hold great promise for revolutionizing ACF applications, particularly in high-resolution soldering tasks such as directly bonding μLED chips onto IC plates.

In summary, we presented a rapid and efficient technique for fabricating pure gold nanoparticle arrays through a positioned self-assembly and laser-irradiated ripening strategy. This technique relied on a layer-by-layer process involving rapid laser-heating, melting fusion, and merging growth, transforming gold colloidosomes into uniform microspheres within minutes. Theoretical simulations revealed that this layer-by-layer phenomenon was due to the finite skin depth of laser, causing a localized photothermal effect on the colloidosome surface. This technique is both straightforward and robust, applicable to various plasmonic nanostructures, even enabling the fabrication of alloy microsphere arrays. Notably, these pure gold microspheres exhibit stable conductivity under deep compression, owing to their superior ductility and malleability compared to conventional polymer-plated gold microspheres. These positioned pure gold microsphere arrays show promise for highly reliable connections in bonding μLED chips onto IC plates, contributing to the advancement of high-resolution display panels.

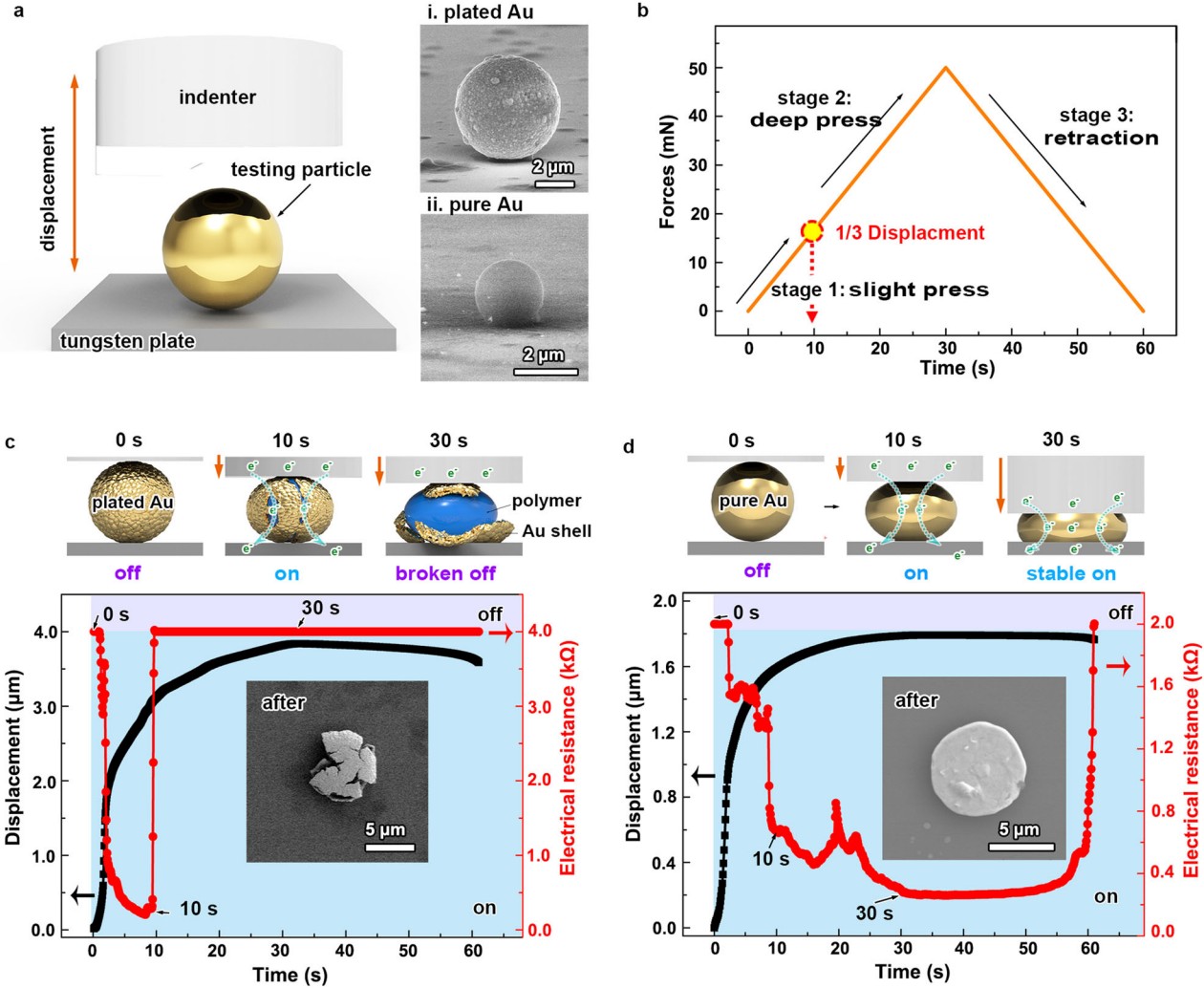

**Fig. 5 | Conductivity under deep compression. a** Schematic illustration of nano-indentation system, insets showing typical SEM images of a polymer-plated gold microsphere (i) and a pure gold microsphere (ii) prepared by our laser irradiation strategy for testing. **b** Force-time curve depicting the displacement program loaded on the indenter for compression. Schematic illustrations and conductivity responses of plated gold microsphere (**c**) and pure gold microsphere (**d**) under deep compression program of (**b**). Insets in (**c**, **d**): corresponding SEM images of a typical plated gold microsphere and a pure gold microsphere after deep compression. Background shading indicates electric "on" (blue) and "off" (purple) states. Source data are provided as a Source Data file.

## Methods

### Materials

Chloroauric acid (HAuCl₄, 99.9%), ethylene glycol (EG, >99.5%), silver nitrate (AgNO₃, >99.8%), 1,4-benzenediol, sodium hydrogen carbonate (NaHCO₃), L-ascorbic acid (AA, >99.7%), Chloroplatinic acid hexahydrate (H₂PtCl₆·6H₂O), N, N-Dimethylacetamide (DMAC, >99.0%) and 1-butanol were obtained from Sinopharm Chemical Reagent Corporation. Poly (diallyldimethylammonium) chloride (PDDA, Mw = 100,000–200,000, 20 wt% in water), polystyrene (PS, average Mw = 35,000), Polydimethylsiloxane (PDMS) and trimethylchlorosilane (TMCS) and Potassium hexachloropalladate (K₂PdCl₆) were purchased from Sigma-Aldrich. Gold-plated conductive microspheres were purchased from Anhui Zhongke Yuanzhen Technology CO.LTD. The deionized water (DI water, 18.2 MΩ cm resistivity at 25 °C) was produced by a Milli-Q integral water purification system. All chemicals and solvents were directly used as received without any further treatment.

### Synthesis of gold nanoparticles

The gold nanoparticles with uniform size were synthesized by using a polyol-reduction method[61]. Briefly, HAucl₄, PDDA, AgNO₃ aqueous solution were added EG in turn, the precursor solution was obtained after shaking evenly. In the precursor solution, the concentrations of HAucl₄, PDDA and AgNO₃ aqueous were 0.0005 M, 0.0025 M and 0.000025 M respectively. Then the precursor solution was placed in an oil bath at 220 °C for 2 h. When the color of the precursor solution changed from the bright yellow into wine-red, it indicated the successful preparation of gold nanoparticles. The gold nanoparticles colloidal solution was centrifuged at 27,975 × g for 20 min after cooling to the room temperature. After repeated centrifugation three times, the collected red precipitated products were re-dispersed in an appropriate volume of DI water and used for further assembly. Gold nanoparticles of different sizes could be obtained by adjusting the pH of the precursor solution before the oil bath according to the previous reports.

### Synthesis of gold octahedrons

Briefly, EG (60 mL), HAuCl₄ (1 M, 30 μL), PDDA (1.2 mL) were added to the glass vial in turn. The mixed solution was stirred vigorously for 3 min at room temperature and immediately put into the oil bath at 220 °C for 2 h without any disturbing. The final products were centrifuged at 27,975 × g for 20 min and rinsed repeatedly with DI water

for three times after cooling to room temperature. Then the final products were dispersed into DI water for subsequent assembly[62].

## Synthesis of gold bipyramids

The gold bipyramids were synthesized by seed growth method[62]. EG (60 mL), HAuCl$_4$ (1 M, 15 μL), PDDA (1.2 mL), AgNO$_3$ aqueous solution (2 M, 48 μL) were added to the glass vial. Subsequently, the mixture solution was put into the oil bath at 220 °C for 4 h. The color of the solution changed from bright yellow to wine red, indicating that Au decahedron seeds of uniform size were successfully prepared. After the gold decahedron seeds were cooled to room temperature, 1,4-benzenediol (0.25 M, 480 μL) and HAuCl$_4$ (0.1 M, 600 μL) were added into the gold decahedron seeds colloid solution. Then the mixture solution was reacted at 50 °C for several hours. The final products were centrifuged at 27,975 × $g$ for three times and re-dispersed into an appropriate volume of the DI water for subsequent assembly.

## Synthesis of Au-Ag-Pt core-shell nanocubes

In a typical process, Au-Ag-Pt core-shell nanocubes were prepared by using Pt$^{6+}$ to selective etching Au-Ag core-shell nanocubes[63]. The specific preparation process is as follows. Firstly, Au-Ag core-shell nanocubes were synthesized by growing Ag nanoshell on the surface of the Au nanosphere seeds. Briefly, NaHCO$_3$, AgNO$_3$ and AA were added to the colloidal solution of gold nanoparticles in the state of agitation, the final concentration was 10 mM, 2 mM and 10 mM, respectively. The evenly mixed solution was placed in the oven at 60 °C for 1 h, and then cooled to room temperature. Secondly, 1 mL of H$_2$PtCl$_6$ (10 mM) aqueous solution was introduced into the Au-Ag core-shell nanocube colloidal solution prepared above and continuously stirred for 1 h at room temperature. After the reaction was completed, the precipitation was centrifuged and dispersed in DI water for the next self-assembly.

## Synthesis of Au-Pd nanoparticles

Au-Pd nanoparticles were obtained by growing Pd nanoparticles on the surface of the Au nanosphere cores. In brief, NaHCO$_3$, K$_2$PdCl$_6$ and AA were added to the colloidal solution of gold nanoparticles in the state of agitation, the final concentration was 10 mM, 2 mM and 10 mM, respectively. The evenly mixed solution was placed in the oven at 60 °C for 12 h. After the reaction was completed, the precipitation was centrifuged and dispersed in DI water for the next self-assembly.

## Preparation of honeycomb PS substrate

The typical process of the preparation of honeycomb PS substrate was divided into the two steps. Firstly, the silicon (Si) wafers of the microhole array (were prepared using a conventional photolithography strategy) were treated with TMCS vapor for 15 min and then used as molds, PDMS precursor (PDMS elastomer and cross linker at a ratio of 10:1) was cast into the Si wafer and put into the oven at 70 °C for 2 h. The silicon wafer mold was carefully peeled off the PDMS substrate after the PDMS was completely dried, then the PDMS mold with a reverse-pattern array was obtained. Secondly, the PS/DMAC solution (5 wt%) was cast into the PDMS mold and dried at 70 °C for 12 h. Finally, the honeycomb PS substrate was successfully prepared after the PDMS mold was removed.

## Preparation of gold colloidosomes

The gold colloidosomes were prepared by using a template-assisted transient emulsion self-assembly strategy[35]. Firstly, 20 μL of 1-butanol was dropped onto the PS film and left to stand for 5 min, this was to make the interior of the PS film completely infiltrated with 1-butanol, then the excess n-butanol was removed by pipette. Secondly, 10 μL of gold nanosphere solution was dropped onto the PS film and left to stand for 5 min, it was to allow the assembly to proceed fully, then the excess gold nanoparticles were rinsed with 1-butanol.

## Preparation of pure gold microspheres

In this experiment, the laser used is a commercially purchased non-focused nanosecond pulse Nd:YAG laser at wavelength of 532 nm with a beam diameter of 4 mm and laser fluence of 22.9 mJ·cm$^{-2}$. The pulse duration is about 30 ns (Fig. 3e), the repetition rate is 10 Hz, and the irradiation time is 10 s. The laser is placed horizontally on an optical platform, the light path is altered through a 45° reflector so that it is vertically incident upon the template substrate (Supplementary Fig. 20). The detailed laser experiment processes are as follow. Firstly, 20 μL of 1-butanol was added to the surface of the PS template with gold colloidosomes and left to stand for 5 min, it was to allow the 1-butanol solution enter the inside of the template fully, then excess 1-butanol on the surface was removed by the pipette. Secondly, the laser fluence and pulse frequency were adjusted to the appropriate value, then the PS template surface was uniformly irradiated vertically for 10 s by slowly moving the template and the pure gold microsphere array was successfully obtained. The optical path system of the laser is used directly and has not been modified. AuAgPtPd alloy microspheres were prepared by the same steps except for replacing Au nanoparticles with Au-Ag-Pt core-shell nanocubes and Au-Pd nanoparticles.

## Conductivity measurements

The single conductive gold microsphere pressure-resistance tests were performed on a Nanoindentation system (Anton Paar, Step 300-NHT3) with a flat punch tungsten carbide tip. In brief, the samples were diluted with ethanol to a suitable concentration, and added onto the washed tungsten substrate. After the samples were dry, the substrate was fixed horizontally to the test fixture. The maximum load of the nanoindentation system was 50 mN, the load rate was 100 mN/min, the contact time was 1 s, the descent speed of the probe was 10,000 nm/min, and the recovery speed was 2000 nm/min.

## Simulation method

Absorption cross section of different gold colloidosomes consisting of different numbers of hexagonal close-packed (hcp) gold nanoparticles with 60 nm in diameter located on the surface of PS were performed using commercially available full-wave finite-difference time-domain software Ansys Lumerical FDTD. PML conditions were assigned to all six planes of the simulation region. Two sufficiently large Total-Field Scattered-Field sources with orthogonal polarizations and phase difference of 90° were used to illuminate the gold colloidosomes located on the PS film. The refractive index of gold was taken from the experimental result of Johnson Christy, and the refractive index of PS and air were set to 1.6 and 1, respectively. The wavelength was set to 530 nm. The absorption cross section was then obtained from the "abs" analysis group.

The heating process of the colloidosomes is simulated in COMSOL 5.5, Multiphysics with the colloidosomes located inside an air cylinder with radius of 2 μm and height of 4 μm embedded in a PS hemisphere with 500 μm in radius. Another hemisphere is assigned to air with the same radius. Constant temperature condition is assigned to the outer surface of the sphere. The heat capacity, mass density and thermal conductivity of gold are set to 129 J/(kg·K), 19,300 kg/m$^3$ and 317 W/(m·K), respectively. The heat capacity, mass density and thermal conductivity of PS are set to 1300 J/(kg·K), 1050 kg/m$^3$ and 0.033 W/(m·K), respectively. The heat capacity of air is set to 1005 J/(kg·K), and its temperature dependent mass density and thermal conductivity are taken from The Engineering Tool Box[64]. The absorption cross section multiplied by the laser intensity gives the total heating power. As the skin depth in gold at 530 nm was about 38 nm[65] which was inferior to the size of each nanosphere (60 nm), it was reasonable to believe that only all the outermost nanoparticles work as the heating source in the experiment. As a result, the total heating power was equally distributed to all the outermost nanoparticles in each case in our heating simulations. The sweeping time step was set to 0.1 ns, in order not to lose any

detail during the heating process. The simulation of colloidosomes on PS immersed in 1-butanol was simulated by simply replacing air by 1-butanol with refractive index of 1.4 used in absorption cross section simulation, and heat capacity 2330 J/(kg·K), mass density 810 kg/m$^3$ and heat conductivity as a function of temperature[66] were used in heating simulation. Domain probes were assigned to seven hcp nanoparticles on the top of the gold colloidosomes, all the heating sources (marked as Union in Fig. 3d), all the cluster, all the media (PS + air or 1-butanol) to monitor their averaged temperature evolution during the heating process. A point probe was purposely set 6 μm far away from the centered gold colloidosomes to monitor the closest neighboring gold colloidosomes heated by the centered one during single-pulse laser illumination, in order to exclude the mutual heating effect[48,67]. Additionally, we observe nonsense values given by this point probe with default calculation accuracy. Any negative value or temperature T at time t ($t > 0$) smaller than T at 0 ns should be attributed to Comsol's calculation accuracy problem.

## Characterizations

The morphology of the final products was characterized by a field emission scanning electron microscope (SEM, SU 8020) and a transmission electron microscopy (TEM, FEI Tecnai G2 F20). The optical photographs of the samples were recorded by a digital camera. The non-focused 532 nm Nd:YAG laser had a pulse duration of 10 ns which was repeated at a frequency of 10 Hz.

## Data availability

All experimental data within the article and its Supplementary Information are available from the corresponding authors upon request. Source data are provided with this paper.

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

## Acknowledgements

The grammar of this article was assisted by ChatGPT. Y.L. acknowledges the financial support from the National Natural Science Foundation of China (Grant Nos. 92263209) and the National Science Fund for Distinguished Young Scholars (Grant No. 51825103). D.L. acknowledges the financial support from the National Natural Science Foundation of China (Grant Nos. 52171232), the Youth Innovation Promotion Association of Chinese Academy of Sciences (Grant Nos. 2022449) and the Special Foundation of President of Hefei Institutes of Physical Science (Nos. BJPY2022B01). A.C. acknowledges the financial support from the Post-doctoral Fellowship Program of CPSF (Grant No. GZC20241743). Y.G. acknowledges the financial support from the Youth Innovation Promotion Association of Chinese Academy of Sciences (Grant Nos. 2020446) and the Plan for Anhui Major Provincial Science & Technology Project (Grant No. 202203a05020003). Y.H.G. acknowledges the financial support from the National Key Research and Development Program of China (2021YFA1401003). Special acknowledgement to Mr. Shixuan Zhao at Department of Physics in City University of Hong Kong and Dr. Feihu Wang at Quantum Science Center of Guangdong-Hong Kong-Macao Greater Bay Area (Guangdong) for their beneficial advice.

## Author contributions

D.L., Y.L., and A.C. conceived the idea. D.L. and A.C. designed the experiments. A.C. carried out the material synthesis and characterizations. Y.F. performed optical and heating simulations. F.Y. and Y.H.G. participated in the discussion. D.L. and A.C. analyzed the data, draw the figures and wrote the manuscript. Y.L., Y.G., and X.T. revised the manuscript, contributed to the discussion, and supervised the project. All authors contributed to the writing of the manuscript. All authors reviewed the manuscript.

## Competing interests

The authors declare no competing interests.
