## [Peer Review File · Nature Communications]

Rapid Fabrication of Gold Microsphere Arrays with Stable Deep-pressing Anisotropic Conductivity for Advanced PackagingIn this manuscript, the authors developed a simple and rapid approach for fabricating pure gold microsphere arrays through a positioned transient-emulsion self-assembly and laser-irradiated ripening strategy. In addition, the layer-by-layer mechanism has been systematically analyzed through theoretical simulations. These designed gold microspheres showed stable conductivity under deep compression, which are promise for ACF applications. However, there are a few limitations explained below, and the authors didn't successfully present the work's novelty.

General comments and queries

1. The novelty of positioned transient-emulsion self-assembly and laser-irradiated ripening strategy to gold microspheres is not convincingly substantiated. In the previous work, the authors have proposed and optimized the emulsion-based template-assisted self-assembly strategy, which can realize high controllability of superstructure size and the applicability to building blocks of a wide range of dimensions, compositions, and morphologies (*Matter*, 2021, 4.3, 927-941.). In addition, the pulsed laser irradiation technology can not only achieve the transformation of nanostructures into nanospheres, but also realize the fabrication of submicro/micro-spheres in existing research (*Scientific Reports*, 2018, 8.1, 11283. *Progress in Materials Science*, 2023, 131, 101004.). The approach and objectives in this manuscript seem to overlap significantly with existing research.

2. The authors described the size distribution of the obtained gold microspheres determined by the templating microhole arrays. In this work, the size of pure gold microspheres obtained is much smaller than these microholes. Moreover, as shown in Figure S5, the distribution of gold microspheres is uneven, and there are both empty microholes and two gold microspheres inside one microhole. Whether these results affect the actual electronic-interconnection applications.

3. The diameter of the self-assembled gold colloidosomes grows with the size of the microhole. Whether the larger gold colloidosomes will spread around and then split into small melted-state droplets during the melting process, and whether the split melted-state droplets can finally be merged into a complete gold microsphere.

4. After the melting and quenching process, will there be stress concentration inside gold microsphere, whether it further affects hardness and electrical conductivity. If so, how can it be eliminated?

Conclusion:

The paper contributes to ongoing research in microsphere synthesis and ACF applications but falls short in several key areas. The claims of novelty and practical applicability lack sufficient evidence and differentiation from existing studies. Regrettably, publish in *Nature Communications* is not recommended.

Reviewer #2 (Remarks to the Author):

The manuscript titled “Rapid and Positioned Fabrication of Pure Gold Microsphere Arrays with Stable Deep-pressing Anisotropic Conductivity for Next-generation Advanced Packaging” the authors demonstrate a newly developed manufacturing method for achieved patterning of gold arrays. The focus of the manuscript is highly explored in the electronic packaging industry and is of interest to the research community specifically in its manufacturing efficiency and scalability. The following recommendations are suggested to the authors before accepting for publication.

1. The figures are beautifully made, but the graphs are a little unreadable. The font size can be increased a little for more legibility.
2. The sentence “However, as the capillary self-assembly is correlated to the delicate evaporation of microsphere suspension, it is easy to cause random particle aggregations on the substrate due to “coffee-ring” effect, making it hard to accomplish long-range accuracy in positioning” is not very convincing. To my acknowledge coffee-ring effect is something observed in the droplet-based non-template method. However, it can be overcome with proper ink formulation in a template-based method (check Professor Kenan Song's group from UGA on template-based micropatterning). Even the citations by the author are from non-template-based methods.
3. The author mentions various temperature regions in the manuscript. Can the author quantitatively mention the regions in line with the figures and content?
4. Even though the manufacturing technique presented here is unique, I am curious to know how it compares to other micro or nano patterning techniques and their scalability such as from Professor Ahmed Busnaina (Northeastern Uni) using 3D printing technique in electronics packaging or Professor Amir Asadi's (Texas A&M Uni) spray deposition method.
5. In the final section the author compared the conductivity of the particle with a commercially available Au-plated particle. Though it is reasonable, their properties can be hugely different since Au plating and the microarray processing techniques are different. Moreover, pure nanoparticles can provide better properties. I suggest the author does some literature studies for a more apple-to-apple comparison. Maybe included in the SI.

Reviewer #3 (Remarks to the Author):

The manuscript “Rapid and Positioned Fabrication of Pure Gold Microsphere Arrays with Stable Deep-pressing Anisotropic Conductivity for Next-generation Advanced Packaging” reports the production of controllable periodically distributed Au microspheres on a patterned polymer substrate for the generation of spatially selective conductive structures. The synthesis and positioning of Au microspheres represents a challenge due to the required micrometric control. In order to address it, the authors propose an already described patterning method to achieve a controllable placement of Au nanoparticles on the polymer substrate and the innovative laser irradiation at low fluences to melt the agglomerated nanoparticles into solid solution Au microspheres. The conductivity of the produced structures is tested under compression conditions, finding that the produced Au microspheres provide a robust mechanical response compared to the commercially available Au coated polymer microspheres. The characterization of the Au microspheres by SEM, and EDX provide a complete overview of both the elemental composition of individual microparticles composed of Au nanoparticles with different morphologies, as well as other alloys. The simulations performed provide a

possible microparticle formation path as a function of the fluence employed.

Overall, the manuscript reports a very interesting and promising laser-based route towards the production of selectively conductive polymer materials suitable for electronics applications. However, some concerns need to be addressed:

- 1) The authors provide a complete explanation of the polymer matrix formation and Au deposition in the patterned holes. However, the details of the laser process are scarce. The exact laser pulse duration, focal spot size, and power employed for irradiation are required for repeatability. Besides, the optical system employed to focus the laser beam and achieve the required fluence and spatial resolution should be described in detail at least in the SI.
- 2) The scanning strategy employed to achieve the self-assembly and laser-ripening process in 1 minute needs to be described, including the technology employed (galvanometric scanners, programmable stage, polygon scanner...) and the experimental details.
- 3) It is mentioned that the maximum size of the gold microspheres is related to the hole volume. However, it is also described in the SI material how the laser fluence limits the maximum Au particle size achievable by laser irradiation. Since the authors do not mention a modification of the fluence employed (22.9 mJ cm^{-2}). Both claims are not compatible unless the fluence was modified in the results presented in Fig. 1f, please clarify it.
- 4) The diameter employed for the model in Fig. 2b is the size of the agglomerated Au nanoparticles that form the colloidosome. The images in Fig. 2a indicate that the starting building blocks are the same Au nanoparticles in every case (Fig. S1), and the size mostly depends on the number of individual nanoparticles present. The authors base their melting model on the size of the agglomerated structures, but a deeper explanation of the effect of the size of the individual nanoparticles conforming them should be provided. This comment is particularly relevant in the case of nonuniform Au sizes in Fig. 2c. Do the fluence values depend on the building blocks size or only on the colloidosomes size?
- 5) To ensure repeatability and facilitate the discussion of the concern raised in the previous comment, please provide the fluence values and laser experimental parameters employed in the different experiments shown in Fig. 2c and 2d.
- 6) The described wavelength selection criteria in chapter 2.4 indicates that plasmonic absorption effects that can maximize the laser energy absorption and so increase the melting process efficiency are expected at 532 nm. The plasmon peak is size and surface dependent. Focusing on the size, the authors employ colloidosomes of 450 nm, 560 nm, and 750 nm. For those sizes the wavelength for maximum absorption would be expected in the red-NIR region. Plasmonic absorption at 532 nm is only found for the initial Au nanoparticles (60 nm building blocks). Hence, the problematic of explaining the melting process from the building blocks size or the colloidosome arises again.
- 7) The evaluated colloidosomes sizes in the simulation (450 nm, 560 nm, and 750 nm) do not match the experimental ones (375 nm, 500 nm, and 800 nm). Please clarify it.
- 8) The laser fluence values used in the simulations are 4.5, 310 13.3 and 17.3 mJ cm^{-2} , while the experimental values shown in Fig 1 are larger. Please clarify it.
- 9) The authors describe the effect of the laser irradiation in 1-butanol and compare it with irradiation in air. To better relate it to the experimental results, a clear description of the colloidosomes situation during irradiation needs to be provided. In principle the authors suggest that 1-butanol is removed before irradiation, but if residuals are left, the authors should provide an estimation of the 1-butanol layer that could remain.
- 10) The tested Au microspheres under compression have a 2 μm size, but this size has not been explored either experimentally or simulated in the previous sections. Please comment the

reason.

11) Suggested text modifications:

- Line 108 “of particle-arrayed ACFs with less than 1 minute”
- Line 139 “In most case, this ripening process follows a layer-by-layer melting and merging mechanism in the morphology evolution,”
- Line 168 “due to the extremely-short time in irradiation,”
- Line 243 “consistent well with our experimental observations”

Responses to Reviewers' Reports

Dear Reviewers

Thank you very much for your valuable suggestions, which are very helpful to improve the quality of our work. We have carefully revised our manuscript according to your critical comments and thoughtful suggestions. All changes made to the text are highlighted in yellow in the revised manuscript. Please see the details below.

Reviewers' comments and our responses:

Reviewer #1

In this manuscript, the authors developed a simple and rapid approach for fabricating pure gold microsphere arrays through a positioned transient-emulsion self-assembly and laser-irradiated ripening strategy. In addition, the layer-by-layer mechanism has been systematically analyzed through theoretical simulations. These designed gold microspheres showed stable conductivity under deep compression, which are promise for ACF applications. However, there are a few limitations explained below, and the authors didn't successfully present the work's novelty.

Comment (1): *The novelty of positioned transient-emulsion self-assembly and laser-irradiated ripening strategy to gold microspheres is not convincingly substantiated. In the previous work, the authors have proposed and optimized the emulsion-based template-assisted self-assembly strategy, which can realize high controllability of superstructure size and the applicability to building blocks of a wide range of*

dimensions, compositions, and morphologies (Matter, 2021, 4.3, 927-941.). In addition, the pulsed laser irradiation technology can not only achieve the transformation of nanostructures into nanospheres, but also realize the fabrication of submicro/microspheres in existing research (Scientific Reports, 2018, 8.1, 11283. Progress in Materials Science, 2023, 131, 101004.). The approach and objectives in this manuscript seem to overlap significantly with existing research.

[Response] Thanks a lot for your valuable comment. We apologize for not clearly clarifying the novelties and significances of our work. Here, we re-elaborate the key points, hoping them more convincing.

Background: As the reviewer mentioned, pulse-laser irradiation is a well-known technique for reshaping and fusing nanostructures into ultra-smooth spheres ranging from nano- to micro-sized scales, utilizing its rapid heating and quenching mechanism ($\sim 10^{-8}$ s). However, **this technique struggles with precise size control**, resulting in a wide size distribution of the irradiated spheres due to random particle fusion, especially at microscale. This issue has challenged scientists for decades, hindering its application in uniform particle synthesis.

Our idea: We proposed that controlling the number of nanoparticles involved in the laser-induced fusion process can address this challenge and achieve uniform-sized microsphere fabrication. In that case, the first step should be able to accurately position a specific number of nanoparticles in a closely-packed state. Our previously-developed templated-assisted transient emulsion self-assembly strategy (Matter, 2021, 4.3, 927-941) can accurately assemble nanoparticles into uniform superparticles (or colloidosomes) and position them precisely. Although robust and well-established, **this strategy still lacks a demonstration in practical applications.**

This work: We combined these two techniques to address the longstanding challenge of micro-sized gold sphere fabrication and positioning. **The novelties** of our approach include:

- 1) **Precise Size control:** Achieving uniform sizes in micro-sized gold sphere synthesis and simultaneously positioning them into arrays.
- 2) **Ideal platform for revealing laser-induced fusion mechanisms:** Providing deeper insights into the interaction between nanosecond-laser pulses and closely-packed colloidosomes quantitatively.
- 3) **Superior conductivity:** Demonstrating that pure gold microspheres offer superior conductive stability under deep compression, ideal for Anisotropic Conductive Film (ACF) applications in advanced electronic packaging.

We believe this work would become a significant step forward in using laser irradiation for the synthesis of uniform metal microspheres, addressing the urgent practical demands of the advanced electronic packaging industry.

Comment (2): The authors described the size distribution of the obtained gold microspheres determined by the templating microhole arrays. In this work, the size of pure gold microspheres obtained is much smaller than these microholes. Moreover, as shown in Figure S5, the distribution of gold microspheres is uneven, and there are both empty microholes and two gold microspheres inside one microhole. Whether these results affect the actual electronic-interconnection applications.

[Response] Thanks a lot for bringing up such excellent question with insightful details.

1). Size much smaller than the microholes:

The much smaller size of the fabricated gold microspheres compared to the templating microholes is due to the limited concentration of gold nanoparticle dispersed in the

water phase. In this work, the saturated concentration of aqueous gold nanoparticles in water phase is approximately 100 nM. At this concentration, only a finite number of gold nanoparticles can be emulsified and assembled into small colloidosomes in the microholes. To obtain larger colloidosomes, the volume of the templating hole needs to be increased.

2). Defects of the microsphere array in microholes:

The SEM image in the Supplementary Figure 5, which shows empty microholes and two microspheres in one hole, was due to the out-of-position phenomenon during laser irradiation. Initially, we did not realize that using a 1-butanol medium could improve positioning accuracy by slowing down the movement of irradiated microspheres. This prevented them from merging into larger ones and forming an uneven size distribution. By reducing this phenomenon and carefully controlling the laser fluence, we achieved a higher filling ratio and more uniform microsphere size, with fewer instances of two microspheres in one hole, as shown in Figure R1 (which also replaces Supplementary Figure 5).

3). Impact on actual applications:

Currently, particle-arrayed ACFs are used instead of traditional randomly-distributed ACFs. This approach ensures reliable conduction with fewer microparticles and reduces the risk of short circuits since the particles rarely move during bonding. In these applications, the smaller gold microspheres in the microholes should have minimal impact.

However, the evolution of the particle-array ACF aims to achieve ultra-fine (microscale) connections, such as one-to-one bonding of micro-sized LED chips. In that case, the smaller microspheres tend to be randomly positioned in the microholes, affecting their

practical application. To address this issue, our next project focuses on control microsphere positioning within the microholes. Our proposed solutions include:

- a. Increasing the depth of the templating microholes to promote the self-assembly and confinement of colloidosomes to their bottom, thereby preserving the order of templating microholes.
- b. Adjusting the hydrophobicity of the microhole. For instance, making the hole bottoms hydrophilic while keeping the walls hydrophobic. This adjustment facilitates the accurate assembly of colloidosomes at the bottom rather than randomly within the microholes.

We believe these steps will allow to achieve a well-positioned array of pure gold microspheres, making them more suitable for practical application in advanced electronic packaging.

Figure R1. Low-magnification SEM image of gold microspheres obtained by laser irradiation.

Comment (3): *The diameter of the self-assembled gold colloidosomes grows with the size of the microhole. Whether the larger gold colloidosomes will spread around and then split into small melted-state droplets during the melting process, and whether the split meltedstate droplets can finally be merged into a complete gold microsphere.*

[Response] This is an excellent question. When ns-laser-pulses melt gold colloidosomes, the resulting gold droplets exist for an extremely brief period (approximately 10^{-6} to 10^{-4} s) before rapidly solidifying. During this short timeframe, spatially-separated droplets find it challenging to move, make contact, and fuse. Therefore, a critical requirement for fusing adjacent gold nanoparticles is that they must be in close contact, such as in a densely-packed state. In our study, gold colloidosomes are composed of densely-packed gold nanoparticles. With an appropriate laser fluence, these gold nanoparticles within colloidosomes preferentially melt and merge layer-by-layer, even as the colloidosome size increases, as shown in Figure R2 (approximately $2.0\ \mu\text{m}$ in diameter). However, at higher laser fluences that cause nanoparticle boiling, colloidosomes tend to fragment into smaller, separated droplets. In such instances, these gold droplets are unlikely to merge unless they maintain contact.

Figure R2. Typical SEM images of $2\text{-}\mu\text{m}$ gold colloidosomes irradiated with different laser fluences.

Comment (4): After the melting and quenching process, will there be stress concentration inside gold microsphere, whether it further affects hardness and electrical conductivity. If so, how can it be eliminated?

[Response] Thanks a lot for this insightful question. It prompted us to revisit our electronic conductivity tests on five randomly-selected pure gold microspheres. Although all microspheres exhibited “stable-on” conductivity, their electronic resistance fluctuated significantly and unpredictably during compression (Figure R3). Initially, we attributed these fluctuations to potential testing errors. However, your question has led us to consider another possibility: these variations may arise from stress concentrations released within the gold microspheres during compression. Exploring the origins and effects of these stress concentrations, and developing strategies to mitigate them, will be important areas for our future research.

Figure R3. The displacement-resistance curves of randomly selected pure gold microspheres.

Reviewer #2

The manuscript titled “Rapid and Positioned Fabrication of Pure Gold Microsphere Arrays with Stable Deep-pressing Anisotropic Conductivity for Next-generation Advanced Packaging” the authors demonstrate a newly developed manufacturing method for achieved patterning of gold arrays. The focus of the manuscript is highly explored in the electronic packaging industry and is of interest to the research community specifically in its manufacturing efficiency and scalability. The following recommendations are suggested to the authors before accepting for publication.

Comment (1): *The figures are beautifully made, but the graphs are a little unreadable. The font size can be increased a little for more legibility.*

[Response] *Thanks a lot for your kind suggestion. We have increased the front size of the graphs in our revised manuscript to improve their legibility. We hope this version makes the figures easier to read.*

Comment (2): *The sentence “However, as the capillary self-assembly is correlated to the delicate evaporation of microsphere suspension, it is easy to cause random particle aggregations on the substrate due to “coffee-ring” effect, making it hard to accomplish long-range accuracy in positioning” is not very convincing. To my acknowledge coffee-ring effect is something observed in the droplet-based non-template method. However, it can be overcome with proper ink formulation in a template-based method (check Professor Kenan Song's group from UGA on template-based micropatterning). Even the citations by the author are from non-template-based methods.*

[Response] We sincerely appreciate this excellent suggestion. As pointed out by the reviewer, the "coffee-ring" effect typically occurs in droplet-based non-template methods but can be mitigated with appropriate ink formulation in template-assisted strategies, as demonstrated by Prof. Song's group (*ACS Nano* 2021, 15, 7, 12057–12068; *Nano Lett.* 2020, 20, 5, 3199–3206). Our original intention was to highlight the challenge of controlling the three-phase (air/liquid/solid) line retraction from the substrate in capillary self-assembly, which often leads to random particle aggregations. Using the term "coffee-ring" effect in this context was inaccurate. Therefore, we have revised the sentence to clarify: “However, as template-assisted capillary self-assembly relies on delicate evaporation of the microsphere suspension, **it is prone to random particle aggregations due to uncontrollable three-phase line retraction, making long-range accuracy in positioning difficult.**” Related references have also been included in our revised manuscript.

Comment (3): The author mentions various temperature regions in the manuscript. Can the author quantitatively mention the regions in line with the figures and content?

[Response] Thank you for this constructive suggestion. The temperature ranges discussed in this study are based on the theoretical melting and boiling points of gold, which are typically 1337 K and 3246 K, respectively. However, at the micro/nano-scale, these temperatures are lower due to size effects, and accurately measuring them is challenging due to the rapid temperature rise (within 30 ns, as depicted in Figure 3f) and the absence of available thermal measurement techniques to accurately characterize the local temperature of nanoparticles (*Light Sci Appl* 2020, 9, 108) during laser irradiation. Therefore, we employed simulation models to elucidate the interaction of laser irradiation with colloidosomes.

Typically, the temperature range critical for transforming gold colloidosomes into larger microspheres corresponds to the melting point region (1337 K), facilitating their fusion. Only under extremely high laser fluence (exceeding $50 \text{ mJ}\cdot\text{cm}^{-2}$), as illustrated in Supplementary Figure 8b, the temperature of gold nanoparticles can reach the boiling point (3246 K), causing them to fragment into smaller particles. Initially, detailed temperature descriptions were omitted from the figures and content. However, we have now incorporated specific temperature data into the revised manuscript to provide clarity. Please refer to page 8, line 4, and page 13, line 1 for more details.

Comment (4): Even though the manufacturing technique presented here is unique, I am curious to know how it compares to other micro or nano patterning techniques and their scalability such as from Professor Ahmed Busnaina (Northeastern Uni) using 3D printing technique in electronics packaging or Professor Amir Asadi's (Texas A&M Uni) spray deposition method.

[Response] Thanks for this interesting question. High-resolution micro/nano patterning techniques in 3D printing, such as two-photon polymerization, projection micro-stereolithography, direct in writing, and electro-hydrodynamic printing, enable the precise fabrication of intricate microstructures with diverse material options. These methods allow for digital modifications to meet specific design needs, making them cutting-edge tools in manufacturing (*Chem. Rev.* 2021, 121, 6246–6291).

Among these techniques, inkjet printing extends direct in writing approaches by manipulating the self-assembly of colloid nanoparticles into hierarchical micro-structured patterns through droplet formation. **However, according to the Young-**

Laplace equation, droplet sizes are typically in the hundred-micrometer range due to ink surface tension, challenging printing resolution improvement.

To overcome this limitation, integrating "top-down" lithography techniques with direct self-assembly strategies has been proposed. For example, Prof. Ahmed Busnaina's group developed a nanoscale 3D printing method based on directed assembly. This technique utilizes hydrophobicity differences between micro/nano-sized patterns and substrates to position and organize nanoparticles into desired structures, beneficial for electronics and biomedical devices (*ACS Nano* 2014, 8, 5, 4547-4558; *Adv. Mater.* 2015, 27, 1759–1766; *ACS Nano* 2017, 11, 8, 7679–7689). **However, constructing these hydrophilicity-difference patterns remains complex, costly, and time-consuming, limiting broader applications.**

Another approach to reducing droplet size to the micrometer scale involves atomization techniques. Prof. Amir Asadi, for instance, utilizes air atomizers to create small droplets from a suspension of water and nanomaterials, which are then sprayed onto substrates to form patterned structures. **Nevertheless, controlling the size distribution of these atomized droplets poses challenges, affecting precision.**

In our previous work (*Matter*, 2021, 4.3, 927-941), we introduced a template-assisted transient emulsion self-assembly strategy. Taking advantage of the partial miscibility of the transient emulsion system, this method allows precise emulsification of the water phase within templating holes. **It facilitates control over droplet sizes down to the submicrometer scale and ensures uniform size distribution.** The process is rapid, robust, and offers high controllability over superstructure self-assembly, applicable to various dimensions, compositions, and morphologies of building blocks. These advantages make it a new approach for manipulating uniform and positioned self-

assembly of nanoparticles. In this study, we further applied this strategy to address longstanding challenges in synthesizing and positioning micro-sized gold spheres by integrating laser irradiation.

Comment (5): In the final section the author compared the conductivity of the particle with a commercially available Au-plated particle. Though it is reasonable, their properties can be hugely different since Au plating and the microarray processing techniques are different. Moreover, pure nanoparticles can provide better properties. I suggest the author does some literature studies for a more apple-to-apple comparison. Maybe included in the SI.

[Response] Thanks a lot for this excellent and constructive suggestion. There are three key reasons we carried out the conductivity test comparison between the gold microspheres and the gold-plated particles:

1. Literature Studies: We reviewed the existing literature on the fabrication of nano- and micro-sized gold spheres, as summarized in Table R1 (also added to our revised Supplementary information). Most works reported the fabrication of uniform gold spheres in nanoscale with fantastic plasmonic optical properties through slow seed growth and etching strategies. For nanoparticle larger than 200 nm, they tend to grow anisotropically into polyhedral structures due to crystalline face energy differentials. Laser irradiation can be used to produce ultra-smooth micro/nano-sized spheres by melting and merging metal particles, but this method struggles to yield uniform sizes. This challenge of fabricating uniform micro-sized and spherical metal particles is longstanding, which limits their particle-arrayed ACF applications. **Thus, there are**

few reports on arranging gold microspheres into non-closely-packed arrays and investigating their conductivity properties.

2. Comparison with polymer microspheres: Polymer particle naturally forms perfect round shapes because of their unique emulsion polymerization processes and can be precisely controlled in size, from nanometers to hundred micrometers. Plating them with a thin metal layer achieves conductivity for industrial ACF applications. However, these core-shell conductive microspheres are unsuitable for soldering tasks, hindering their **ACF applications extended to bonding electronic components. A feasible solution is to replacing them with pure metal microspheres,** as highlighted by *Dexermials*, a Japanese company specializing in advanced materials and components for electronics (<https://techtimes.dexerials.jp/en/bonding/secret-behind-paf/>). Therefore, direct comparing the conductivity between the gold-plated microspheres with pure gold microspheres is essential for testing potential ACF applications. With these reasons, we think it would be a kind of “apple-to-apple” comparison.

3. Practical application: Initially, particle-arrayed ACFs aimed to replace traditional randomly-arranged ACFs, ensuing stable area conductivity in bonding circuits typically a few hundred-micrometers wide. In this case, comparing the area conductivity of pure-gold-microsphere-array ACF with gold-plated one shows no significant advantage, as the area conductivity remains stable even if some gold-plated spheres were “broken off”. However, for ultra-fine bonding connection (e.g., one-to-one microscale connections), the advantages of pure gold microspheres, particularly their conductivity under deep impression, become apparent and superior. **Therefore, directly comparing the fundamental conductivity properties of gold-plated and pure gold microspheres is essential.** Our results reported in this work indicate that our pure gold

microspheres offer superior conductivity performance, validating their potential for advanced electronic packaging.

Table R1 Literature studies of the fabrication of nano- and micro-sized gold spheres.

Methods	Sizes (μm)	CV	State	References
seed growth	0.005-0.15	4.8%	random	Part Part Syst Charact 31 , 266-273 (2014)
Multi-step growth-etching	0.1-0.2	/	random	ACS Nano 7 , 12, 11064-11070 (2013)
pulsed UV laser treatment	0.02-2	/	random	Sci Rep 8 , 11283 (2018).
pulsed laser melting in liquid	0.05-0.25	/	random	J Phys Chem C 122 , 21659–21666 (2018)
electroless plating	1.9	5%	random	Adv Funct Mater 17 , 618-622 (2017)
thermal annealing	0.15-0.4	/	array	Adv Sci 11 , 2306239 (2024)
One-pot synthesis	0.06-0.17	/	random	Chem Mater 33 , 7, 2593-2603 (2021)
transient emulsion self-assembly followed with laser irradiation	0.2-2	4.03%	array	This work

Reviewer #3

The manuscript “Rapid and Positioned Fabrication of Pure Gold Microsphere Arrays with Stable Deep-pressing Anisotropic Conductivity for Next-generation Advanced Packaging” reports the production of controllable periodically distributed Au microspheres on a patterned polymer substrate for the generation of spatially selective conductive structures. The synthesis and positioning of Au microspheres

represents a challenge due to the required micrometric control. In order to address it, the authors propose an already described patterning method to achieve a controllable placement of Au nanoparticles on the polymer substrate and the innovative laser irradiation at low fluences to melt the agglomerated nanoparticles into solid solution Au microspheres. The conductivity of the produced structures is tested under compression conditions, finding that the produced Au microspheres provide a robust mechanical response compared to the commercially available Au coated polymer microspheres.

The characterization of the Au microspheres by SEM, and EDX provide a complete overview of both the elemental composition of individual microparticles composed of Au nanoparticles with different morphologies, as well as other alloys. The simulations performed provide a possible microparticle formation path as a function of the fluence employed.

Overall, the manuscript reports a very interesting and promising laser-based route towards the production of selectively conductive polymer materials suitable for electronics applications. However, some concerns need to be addressed:

Comment (1): *The authors provide a complete explanation of the polymer matrix formation and Au deposition in the patterned holes. However, the details of the laser process are scarce. The exact laser pulse duration, focal spot size, and power employed for irradiation are required for repeatability. Besides, the optical system employed to focus the laser beam and achieve the required fluence and spatial resolution should be described in detail at least in the SI.*

[Response] *Thanks for your thoughtful suggestion. We have incorporated additional details of the laser experiment into the revised manuscript. Specifically, the laser employed in this works was Nd:YAG laser in non-focused state, with 15 ns in pulse*

width, 5 mm in spot diameter, 10 Hz in repetition frequency and tunable fluences ranging from 0-50 mJ·cm⁻². For the optical system setup was presented in Supplementary Figure 20, the laser was emitted horizontally, passes through a 45-degree total reflection mirror, and then vertically incident on the sample.

***Comment (2):** The scanning strategy employed to achieve the self-assembly and laser-ripening process in 1 minute needs to be described, including the technology employed (galvanometric scanners, programmable stage, polygon scanner...) and the experimental details.*

[Response] We greatly appreciate your construction suggestion. Currently, our scanning strategy involves manually moving the substrate (1×1 cm²) to cover the entire area for laser irradiation (5 mm in spot size). This method is feasible because, once the colloidosomes are fully transformed into ultra-smooth gold spheres under appropriate laser fluences, their morphology remains stable even with additional laser pulses. This stability allows us to manually move the substrate around to achieve uniform laser spot irradiation across the substrate, ensuring consistent morphology transformation of the colloidosomes.

However, this manual approach is suitable only for laboratory-scale experiments to validate our concept of template-assisted transient emulsion self-assembly combined with laser irradiation. As the reviewer suggested, for more efficient and large-scale fabrication to meet industrial needs, programmable manipulation for substrate scanning is necessary. We believe this transition will be manageable by simply set the scanning step at 5 mm.

Comment (3): *It is mentioned that the maximum size of the gold microspheres is related to the hole volume. However, it is also described in the SI material how the laser fluence limits the maximum Au particle size achievable by laser irradiation. Since the authors do not mention a modification of the fluence employed (22.9 mJ cm^{-2}). Both claims are not compatible unless the fluence was modified in the results presented in Fig. 1f, please clarify it.*

[Response] Many thanks for bringing this point to our attention. As noted by the reviewer, we can increase the size of colloidosomes by raising the gold nanoparticle concentration when the templating hole volume is fixed. In this case, we maintained a laser fluence of $22.9 \text{ mJ}\cdot\text{cm}^{-2}$ to ensure all colloidosomes, regardless of size, transformed into microspheres without fragmenting into smaller nanoparticles (blue region in Figure 1f). Once the gold nanoparticle concentration reached saturation at about 100 nM, we enlarged the templating volume by approximately 15 times, allowing the colloidosomes to reach 2.5 micrometers (yellow region in Figure 1f). Consequently, we adjusted the laser fluence to $39.1 \text{ mJ}\cdot\text{cm}^{-2}$, as larger microspheres require more energy to melt and merge. We apologize for not including this modification in the manuscript; the relevant information has now been added on page 10, line 5.

Comment (4): *The diameter employed for the model in Fig. 2b is the size of the agglomerated Au nanoparticles that form the colloidosome. The images in Fig. 2a indicate that the starting building blocks are the same Au nanoparticles in every case (Fig. S1), and the size mostly depends on the number of individual nanoparticles present. The authors base their melting model on the size of the agglomerated structures, but a deeper explanation of the effect of the size of the individual nanoparticles*

conforming them should be provided. This comment is particularly relevant in the case of nonuniform Au sizes in Fig. 2c. Do the fluence values depend on the building blocks size or only on the colloidosomes size?

[Response] Thanks for your excellent question. Exactly, the building blocks size and the colloidosomes size both have effects on the choice of the fluences. Specifically, the size of building blocks determines the minimum laser fluence required to start the melting-merging process. Whereas the size of colloidosomes determines the minimum laser fluence required to complete the transformation process which has been indicated in Figure 2b. When changing the size of building block from 60 nm to 120 nm, slightly lower laser fluence is required to start melting process but slightly higher laser fluence is required to start the transformation process. We start the study from changing the size of building block of *hcp* packed colloidosome from 60nm to 120nm, as seen in Figure R4a-b and select the outmost building blocks on the top surface as heat sources (Figure R4c). As we have removed all the building blocks whose surface touches the sphere of 684 nm in diameter, the true dimension of colloidosomes composed of building blocks with 120 nm in diameter is smaller (as seen in Figure R3b), while the colloidosomes composed of building blocks with 60 nm in diameter is closer to a true sphere (as seen in Figure R3a).

Figure R4. Changing building blocks of 60 nm in diameter (a) to 120 nm in diameter (b) and the selected building blocks used as heating sources in thermal simulation (c).

As can be read from Figure R5a, the absorption and scattering cross sections of the colloidosomes with large building block share the similar features as those shown in Figure 3a in the manuscript, where the absorption cross section slightly decreases after 550 nm and rise again in the near-NIR domain, and the scattering cross section remains stably low until 550 nm and then rapidly augments. This can be attributed to the intrinsic absorption around 530 nm of gold and higher-order resonant in the near-NIR domain. The near-field electric field distribution shows strong electric field of the outermost gold nanospheres, while the inner nanospheres remain untouched by light, as revealed by Figure R5b. The biggest temperature rise of the gold nanospheres is ~1200 K, in contrast to 1076 K for the smaller-building-block counterpart, as seen in Figure 3f. This more than 10% of temperature rise with smaller absorbed optical energy may be attributed to the more compact distribution of Union for heating (meaning that the illuminated larger building blocks on the top surface distribute in a smaller region and each one is quasi-isothermal, while the illuminated smaller building blocks distribute in a larger area and interfaced by air) and the collective heating effect [ACS Nano 2013, 7, 6478-6488]. This leads to the slightly decreased threshold of laser fluence for melting. However, we compare the thermal behaviors of the central gold nanospheres inside the colloidosome, which shows slightly lower temperature rise of central building block of 120 nm than 60 nm and this may probably be caused by its smaller absorption cross section. The more rapid temperature rises of former result from the smaller distance between center to lower-surface distance of illuminated building blocks (gold structures are quasi-isothermal). Thus, a stronger laser fluence should be required to transform building blocks of 120nm to gold nanospheres than that of 60 nm.

Figure R5. (a) Absorption, scattering and extinction cross sections of gold colloidosome $D=684 \text{ nm}$ composed of nanospheres with 120 nm in diameter. (b) Its near-field distribution of electric field at 530 nm . (c) Thermal distribution of the colloidosome when union achieves the most elevated temperature. (d) Temperature evolution of the central sphere of the colloidosome composed of nanospheres with 120 nm in diameter (red curve) in comparison to that composed of nanospheres with 60 nm in diameter (blue curve).

To experimentally prove this point, we used the gold nanospheres with the larger diameter (about 120 nm , Figure R6a and 5b) as building blocks, and assembled gold colloidosome array successfully as shown in the Figure R5c. When the laser fluence was $2.1 \text{ mJ}\cdot\text{cm}^{-2}$, the surface of gold colloidosome with 120 nm nanospheres began to melt slightly. By contrast, the gold colloidosome with 60 nm nanospheres remained

intact at the same fluence (Figure 2a). As the laser fluence was gradually increased into $22.9 \text{ mJ}\cdot\text{cm}^{-2}$ (which was sufficient), the gold colloidosome were gradually transformed into gold microsphere (Figure R6d-h). The above experimental results are in good agreement with the simulation results.

Figure R6. (a) TEM image of 120 nm gold nanospheres. (b) The histogram of size statistics of the 120 nm gold nanospheres. (c)-(h) Typical SEM images of gold colloidosomes assembled with 120 nm gold nanospheres irradiated under varied laser fluences.

Comment (5): To ensure repeatability and facilitate the discussion of the concern raised in the previous comment, please provide the fluence values and laser experimental parameters employed in the different experiments shown in Fig. 2c and 2d.

[Response] Thanks a lot for your valuable suggestions. The experiments shown in Figure 2c and 2d were conducted with a laser fluence of $22.9 \text{ mJ}\cdot\text{cm}^{-2}$, a frequency of 10 Hz and a duration of 10 seconds. These parameters ensure the complete transformation of colloidosomes smaller than $1 \mu\text{m}$ into microsphere, regardless of the sizes and shapes of the building blocks. We apologize for not clarifying this in the original manuscript. This point has now been highlighted in the revised manuscript on page 14, line 17.

Comment (6): *The described wavelength selection criteria in chapter 2.4 indicates that plasmonic absorption effects that can maximize the laser energy absorption and so increase the melting process efficiency are expected at 532 nm. The plasmon peak is size and surface dependent. Focusing on the size, the authors employ colloidosomes of 450 nm, 560 nm, and 750 nm. For those sizes the wavelength for maximum absorption would be expected in the red-NIR region. Plasmonic absorption at 532 nm is only found for the initial Au nanoparticles (60 nm building blocks). Hence, the problematic of explaining the melting process from the building blocks size or the colloidosome arises again.*

[Response] Thanks a lot for raising this question. First of all, we agree with referee that the plasmon peak is size and morphology dependent, and the resonance absorption peak of gold nanosphere is only found around 532 nm when its diameter $\leq 60 \text{ nm}$, while the resonant peak will be redshifted when the size of gold nanosphere gets larger. However, the fact that we should point out here is that the above-mentioned statement is true for Mie resonant structures such as spheres, ellipsoids. Any structures whose modes cannot be decomposed into spherical series are not Mie resonant structures such as cube, dimer,

trimer, etc. Obviously, the colloidosomes employed in this experiment are not Mie resonant structure and its absorption/scattering spectrum are largely broadened due to multiple scattering which has been observed in several previous works in the literature [ACS Nano 2018, 12, 2643-2651; ACS Nano 2022, 16, 910-920; Science 2010, 328, 1135-1138; Science 2003, 302, 419-422]. The absorption curves of different colloidosomes given by Figure 3a do not preclude the possibilities of existing absorption peaks in the NIR region. But they surely confirm the reasonable choice of laser 532 nm as light source for plasmonic heating of different colloidosomes due to their common absorption peaks around 532 nm which are consistent with observations in [ACS Nano 2018, 12, 2643-2651; ACS Nano 2022, 16, 910-920]. In our reply to Question 4, we find the absorption cross section of $4.36 \times 10^5 \text{ nm}^2$ with superparticle $D=684 \text{ nm}$ composed of building blocks with 120 nm in diameter, in contrast to $4.64 \times 10^5 \text{ nm}^2$ for the superparticle $D=684 \text{ nm}$ composed of building blocks with 60 nm in diameter, which make a difference of 6%. This intrinsic high absorption around 532 nm allows to finish the gold microsphere within 1 min without changing laser, thus ensuring the cost-effectiveness and time-saving of our fabrication method.

Comment (7): The evaluated colloidosomes sizes in the simulation (450 nm, 560 nm, and 750 nm) do not match the experimental ones (375 nm, 500 nm, and 800 nm). Please clarify it.

[Response] Thank you for pointing out this discrepancy. We are sorry to make readers confused about the discrepancy between models used in the simulation and those studied in the experiment. First of all, we should point out that neither the dimensions of sphere-like colloidosomes studied in the experiment nor those created by Matlab in

a well-ordered *hcp* form are accurate values. On one hand, the uncertainty of existence of one nanoparticle in the SEM measurement would have resulted in an uncertainty of 60 nm when determining the dimension of colloidosome. On the other hand, as can be seen in our response to **Comment (4)**, all building blocks touching the colloidosome's frontier are removed when calculating the coordinate of each building block in Matlab. This could cause a maximum uncertainty of 120 nm (two nanospheres touching frontiers at each side) in dimensions along *x*, *y* or *z* axes. To improve the accuracy of the nominal diameter of colloidosomes in the manuscript, we measure their dimensions in *x,y,z* directions, which read (432 nm, 400 nm, 364 nm) for the smallest one, (556 nm, 507 nm, 465 nm) for the second one and (680 nm, 704 nm, 667 nm) for the biggest one, respectively, resulting in averaged diameters of 399 nm, 509 nm and 684 nm by averaging the dimensions in *x,y,z* directions. We have updated these values in the revised manuscript. Besides, as the diameters of modelled colloidosomes, i.e., 399 nm, 509 nm and 684 nm fell within the experimental range, we proceeded with the simulations without adjustments. The results show that this discrepancy minimally affected the initial melting process of the surface-layered nanoparticles (Figure 3f).

Comment (8): *The laser fluence values used in the simulations are 4.5, 13.3 and 17.3 mJ cm⁻², while the experimental values shown in Fig 1 are larger. Please clarify it.*

[Response] We thank the Referee very much for pointing out this discrepancy and helping us find this severe mistake when modelling laser induced heating of colloidosome of 399 nm in diameter, which should be 9.2 mJ·cm⁻² rather than 4.5 mJ·cm⁻². We should apologize for this carelessness and we have revised the relevant errors throughout the manuscript and the Supplementary Information. We have also

double-checked all the values declared in this manuscript to ensure that no such mistake exists in our revised version. In our Response to *Comment (8)*, the averaged diameter of 399 nm and 509 nm is in close proximity to the measured diameters of 375 nm and 500 nm in experiment. As to the modeled one of 684 nm in diameter, our simulation result revealed a merging threshold of $17.3 \text{ mJ}\cdot\text{cm}^{-2}$ which lies in between the thresholds for 800 nm and 500 nm. To further demonstrate the reliability of our simulation results, we can define a variable derived from equation (11) in the Supplementary Information to evaluate the discrepancy between calculated threshold with measured one: $R(d_p) = \frac{d_p^3}{(K_{absorption} \times d_p - a) \cdot J(d_p)}$. Taking $d_p=399, 509, 684, 800$ and $J(d_p)=9.2, 13.3, 17.3$ and 22.9 into $R(d_p)$, we can get the relation as follows: $R(399) : R(509) : R(684) : R(800) = 1 : 0.87 : 1 : 0.96$, demonstrating a reasonable threshold of $17.3 \text{ mJ}\cdot\text{cm}^{-2}$ for colloidosome of 684 nm in diameter.

In addition, we should emphasize that our findings reported in this manuscript demonstrate a simple yet effective approach for fabricating pure gold microsphere arrays through a positioned transient-emulsion self-assembly and laser-irradiated ripening strategy for ACF application. And our simulations provide a quantitative method to calculate the plasmon induced photoabsorption-thermodynamics of gold colloidosomes when irradiated by laser at 532 nm, revealing the colloidosome dimension dependent merging threshold and helping understand the melting-merging mechanism. In this sense, the true dimensions of colloidosomes or the merging thresholds with 100% accuracy are not the most important things to consider for our above objective.

Last but not least, we should express our gratitude again to the Referee to help us avoid this severe error in our manuscript.

Comment (9): The authors describe the effect of the laser irradiation in 1-butanol and compare it with irradiation in air. To better relate it to the experimental results, a clear description of the colloidosomes situation during irradiation needs to be provided. In principle the authors suggest that 1-butanol is removed before irradiation, but if residuals are left, the authors should provide an estimation of the 1-butanol layer that could remain.

[Response] Thanks for your insightful suggestion. The boiling temperature of 1-butanol is 390.8 K at 1 atm. On one hand, to ensure complete evaporation of the 1-butanol in the microhole, we placed the templates in an oven at 368 K for 1 hour before the air comparison experiments. On the other hand, as demonstrated by Figure 4a, the surface temperature of colloidosomes can still reach ~520 K during laser irradiation although the whole 1-butanol remains almost unchanged. This indicates that residual 1-butanol layer that could be heated up to, or even beyond its boiling temperature and thus, this layer would be evaporated during the first pulses irradiation. When reduce the 1-butanol layer to several tens of nanometers, e.g., D=450 nm incorporating the whole colloidosome, the temperature of the topmost gold spheres can reach much higher temperature (> 800 K, see Figure R7) as compared to that in the whole 1-butanol environment (Figure 4a) due to the much lower thermal conductivity of air. This temperature is far beyond the boiling temperature of 1-butanol. Thus, we believe that there should be no residual 1-butanol layer after baking in oven, or the residual 1-butanol layer would be evaporated during the first pulses irradiation based on our thermal simulation and this would not affect the gold microsphere synthesis, ACF application or the mechanical conclusion.

Figure R7. Heating of gold colloidosome with 1-butanol layer of 399 nm in diameter.

Comment (10): The tested Au microspheres under compression have a 2 μm size, but this size has not been explored either experimentally or simulated in the previous sections. Please comment the reason.

[Response] Thanks a lot for your constructive question. Actually, the experimental results for fabricating 2 μm gold microspheres were plotted in Figure 1f (yellow region), obtained by enlarging the templating hole volume 15 times and increasing the laser fluence (Figure R8). Additional experimental results, including SEM images, has been also added to the revised supporting information (Supplementary Figure 19).

Figure R8. (a) Typical SEM image of 2.5 μm gold colloidosomes. (b) Typical SEM image of a 2.0 μm gold microsphere.

Comment (II): Suggested text modifications:

- Line 108 “of particle-arrayed ACFs with less than 1 minute”
- Line 139 “In most case, this ripening process follows a layer-by-layer melting and merging mechanism in the morphology evolution,”
- Line 168 “due to the extremely-short time in irradiation,”
- Line 243 “consistent well with our experimental observations”

[Response] Thanks a lot for these excellent suggestions. We have incorporated these text modification changes in the revised manuscript. The updated sentences are “of particle-arrayed ACFs **within 1 minute**”, “In most case, **the** ripening process follows...,” “due to the **short** time in irradiation (**~ 30 ns**)” and “consistent well with **experimental observations**”. Additionally, we have double-checked the grammar and expression throughout the remaining text to enhance the readability of the manuscript.

<End>

Reviewer #1 (Remarks to the Author):

All the revisions have been checked and I recommend this manuscript for publication now.

Reviewer #2 (Remarks to the Author):

The author has done good task addressing all the comments and making necessary changes to the manuscript and the supporting information. The manuscript can be accepted in its current form for publication.

Reviewer #3 (Remarks to the Author):

The authors have fully answered my comments and made the required modifications regarding the fluence values and differences between the experiments and the simulations. Hence, the manuscript can be published in its current form.